# A smartphone analogy to explore the origin of animals

Iñaki Ruiz-Trillo [1,2,✉], Elena Casacuberta[1], Nicholas H Brown[1,3] & Ricard Solé[1,2,4]

## Abstract

**How animals evolved from their unicellular ancestor is a fundamental biological question. The fact that all animals are monophyletic—sharing a single common ancestor—implies their origin from unicellular eukaryotes was likely driven by rare and highly advantageous innovations. While the fossil record and initial genomic comparisons suggested animals originated by the rapid acquisition of many novel genes, new research on animal's closest unicellular relatives reveals most of those genes originated before animals evolved. Here we present a new model for animal origins, which shares similarities with the origin of one of the greatest technological innovations of our time: the smartphone. We show that the origin of both animals and smartphones was due to the integration and repurposing of pre-existing components driven by a novel "operating system", rather than the sudden emergence of many new parts. This model offers testable predictions and a new theoretical framework for understanding complex biological innovation.**

**Keywords** Multicellularity; Evolutionary Transition; Technological Evolution; Metazoans; Smartphone
**Subject Category** Evolution & Ecology

## Introduction

The diversity of animal life, encompassing both extant and extinct forms, is astounding, yet all animals share a common ancestor (Haeckel, 1874; Wainright et al, 1993; Medina et al, 2003; Ruiz-Trillo et al, 2008; Rokas, 2008; Torruella et al, 2012; Ros-Rocher et al, 2021). This suggests that at some point in evolution, a single-celled ancestor of animals acquired some innovation -that is, a novel trait or capacity that created new functional or ecological possibilities (Erwin, 2015)—that subsequently enabled the formation of the first animal. The fact that there is currently only one lineage of animals, rather than several parallel ones, suggests that the emergence of extant animals was a rare event, most likely involving some degree of contingency. This evolutionary step has

some similarity to the evolution of eukaryotes from prokaryotic ancestors, since both of these events are unique: there remains just one single lineage of descendants, which were very successful. Note, however, that there have been other acquisitions to multicellularity in eukaryotes that are independent of the one that gave rise to animals, such as the evolution of land plants, multicellular fungi, or different types of algae (Ruiz-Trillo et al, 2007; Knoll, 2011; Sebé-Pedrós et al, 2017).

What were the key innovations, if any, that eventually gave rise to the first animal? This is the question we and others have been seeking to answer by characterizing the closest living unicellular relatives to animals, many of which have been discovered and studied quite recently (reviewed in Ros-Rocher et al, 2021). Interestingly, the more we have studied animal's relatives, the more complex the unicellular ancestor appears to be, whose genome already had many genes that are crucial for animal multicellularity. A few genes (estimated at just 25; Paps and Holland, 2018)) are still, apparently, currently unique to animals—a number expected to decrease with more unicellular relative sequencing. One possibility is that one or more of these remaining animal-specific genes was, in fact, the key "hard" innovation that led to the emergence of animals. Here, however, we explore an alternative view: that the innovation was not a specific gene or pathway but a new 'operating system' that controlled the diverse functions already in place, and allowed for new combinations of those.

To explore this alternative view, we begin by reviewing current models for the origin of animals and examining how recent evidence challenges them. We then turn to a powerful, if sometimes misunderstood, tool of scientific inquiry: the analogy. Analogies in science are not rhetorical flourishes but instruments of discovery-they can spark new hypotheses by revealing unseen connections between domains. For an analogy to be scientifically fruitful, it must have three components (Hesse, 1966): the *positive analogy* (known similarities that ground the comparison), the *negative analogy* (known differences that define its limits), and, most importantly, the *neutral analogy* (the unknown properties where the comparison might or might not hold). It is by exploring this "neutral analogy" that new, testable hypotheses are generated (Hesse, 1966). We here use the evolution of the smartphone as an analogy for the origin of animals. We invite the reader to engage with this comparison in the spirit just outlined: to recognize and/or challenge the positive analogies, to acknowledge the negative ones, and to challenge and

[1]Institut de Biologia Evolutiva (CSIC-Universitat Pompeu Fabra), Barcelona, Spain. [2]Institució Catalana de Recerca i Estudis Avançats (ICREA), Barcelona, Spain. [3]Department of Physiology, Development and Neuroscience, University of Cambridge, Cambridge CB2 3DY, UK. [4]Santa Fe Institute, Santa Fe, NM 87501, USA.
✉E-mail: inaki.ruiz@ibe.upf-csic.es

explore the neutral ones that may inspire fresh thinking. Through this metaphor, our goal is not to provide a pedagogical explanation about animal origins but rather to challenge old assumptions, generate new questions, and guide future research endeavors about one of life's major evolutionary transitions.

## Initial models of the origin of animal

Discussions about the origin of animals have centered on three core concepts. These are: (1) the explosive origin of animals in the Cambrian; (2) the apparent simplicity of unicellular eukaryotes compared to animals, and (3) the expectation of stepwise evolution from simpler unicellular eukaryotes to simpler animals, and then to more complex animals. We now consider these three concepts, which have proved to be misleading.

The idea that animals arose due to an incredible burst of genetic innovation was primarily spurred by the fossil record (Erwin et al, 2011; Erwin and Valentine, 2013; Briggs, 2015). Notably, the diversity of animal life suddenly appears in the fossil record over a relatively short span of approximately 20 to 25 million years, at the beginning of the Cambrian period, 540 million years ago. Cambrian fossils are beautifully preserved and show diverse, complex animal-like body plans. Representatives of most animal clades are found, including annelids (worms), arthropods (insects, crabs), molluscs (snails, octopuses), and chordates (including humans and fish) (Fedonkin, 2007). There are even fossils from the Cambrian that cannot be attributed to any living animal, implying a high degree of biological innovation and diversification during that period. This phenomenon, known as the "Cambrian explosion", led to the concept that the origin of animals itself involved an explosion of genetic innovation (Gould, 1989; Conway Morris, 2003; Powell, 2012).

This concept of the explosive origin of animals has been strengthened by the apparent sparsity of fossils in the pre-Cambrian Ediacaran period that could be related to living animals or Cambrian fossils. However, to counter this view, some have argued that the Cambrian explosion is an artifact of the fossil record, and similar animals could have originated long before but remain undetected because they did not fossilize well, probably due to their small size or the lack of hard body parts (Erwin and Valentine, 2013). Moreover, recent analyses of the fossil record and molecular clock estimates -which consistently place the origin of Metazoa nearly 200 million years before the Cambrian- have further challenged the explosive origin of animals, implying a long history of cryptic evolution (Erwin, 2020; Wood et al, 2019). Instead, authors have proposed that the early metazoan diversification was indeed the result of successive, transitional radiations that originated in the late Ediacaran, with the Cambrian explosion itself representing a later radiation of crown-group bilaterians (Wood et al, 2019). Thus, the concept of the "Cambrian explosion" has been reinterpreted. It is now seen not as an explosion of genetic innovation, but as a rapid diversification of bilaterian body plans. This radiation was likely fueled by a combination of factors, including environmental changes (e.g., rising oxygen levels), new ecological pressures (e.g., a predator-prey "arms-race"), and, probably, the co-option of ancient gene regulatory networks into new roles for body plan patterning (Erwin and Valentine, 2013).

Regarding the second concept (i.e., the apparent simplicity of unicellular eukaryotes compared to animals), our understanding of animal origins has been influenced by a human-centric perspective on protists, manifesting in two ways. First, we tend to overstate the importance of animals (and plants and fungi) within the broader context of eukaryotic diversity. Animals or metazoans are undoubtedly remarkable organisms and, clearly, their emergence profoundly changed environments at a global scale (Erwin and Tweedt, 2012; Cribb et al, 2019; Gougeon et al, 2025). However, the reality is that animals constitute only one of approximately 100 major, deep-branching eukaryotic lineages (del Campo et al, 2014). Second, we tend to view animals as the pinnacle of evolutionary complexity while mistakenly considering unicellular organisms as simple. This perception arose partly because, for many years, yeasts served as model organism representatives of protists for the molecular biology field, and they are indeed much simpler than animals, in both genomic gene content and morphological complexity. This reinforced the concept that the evolution of animals from unicellular ancestors involved a great increase in genetic complexity, as suggested by the Cambrian explosion. However, we now know that this is not the case, as it turns out that yeasts have discarded numerous genes; the early branching chytrid fungi have many more genes in common with animals than yeasts, as does *Dictyostelium* and other protists shown in Fig. 1. Unicellular organisms closely related to animals not only share many genes with animals but can also exhibit complex cellular behaviors and non-linear life cycles which include temporal multicellular structures (Ros-Rocher et al, 2021; Ruiz-Trillo et al, 2023). We can now confidently infer that unicellular organisms are far from simple and that the unicellular organisms that gave rise to animals were much more complex than previously thought.

Finally, there is the expectation of a stepwise evolution from simpler unicellular organisms to simpler animals, and then to more complex animals. This stepwise progression is central to the two most popular models for the origin of animals: the choanoblastea model and the synzoospore model. The choanoblastea model, for instance, envisions a progression resembling current living organisms: from a unicellular ancestor similar to living choanoflagellates, to the first animal resembling a living sponge, and then to the rest of the animals (Nielsen, 2008; Nielsen, 2023; Brunet and King, 2017). This model therefore has a straightforward linear progression of transformation between "living species". However, we identify three problems with this model. First, it posits that animals descended from a choanoflagellate-like ancestor. The finding that choanoflagellates branched off the closest to the emergence of animals does not mean that animals descended from choanoflagellates. Both animals and choanoflagellates share a common ancestor that was neither a modern choanoflagellate nor a modern animal. Living choanoflagellates have evolved for the same length of time as humans have from their ancient ancestor, so that they may differ from the ancestor just as substantially. In addition, we now know of three other unicellular relatives of animals (corallochytreans, ichthyosporeans, filastereans, Figs. 1 and 2), which are morphologically very different from choanoflagellates (Fig. 2) and which share genes with animals that are missing in choanoflagellates. Indeed, given the time span since their shared ancestors, it is impossible to definitively state whether any of these protists closely resemble the common ancestor. Second, this model also assumes that the first animal was similar to living sponges.

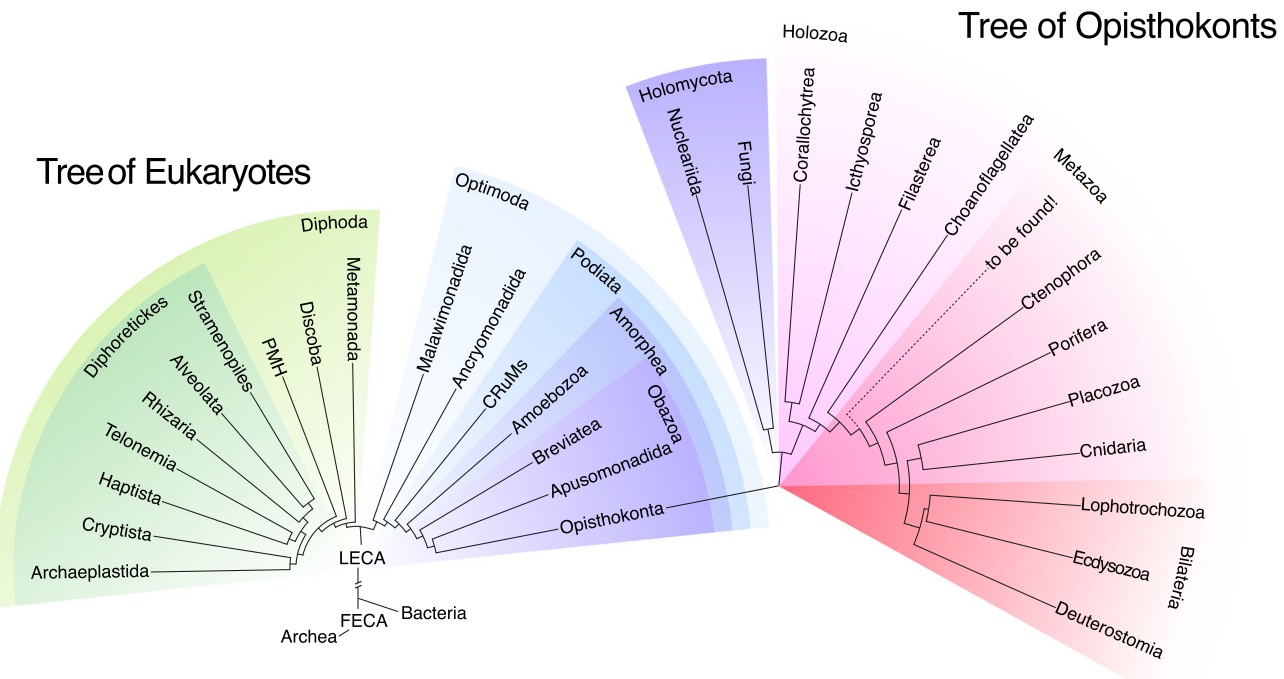

**Figure 1.  A schematic tree of eukaryotes (left) and Opisthokonta (right).**

Relationships and distances based on Torruella et al, 2025 and Liu et al, 2024).

Although sponges may possess one of the simplest multicellular structures among extant animals, we lack definitive evidence that the first animal resembled them. Indeed, this is quite unlikely given the time span that has occurred since they last diverged from other animals. Thus, attempting to model the origin of animals based on living organisms results in an unnecessary restriction. Third, the choanoblastea model was in part based on the perceived homology between a distinctive and evolutionarily conserved cellular structure called the collar complex, shared by all choanoflagellates and the choanocyte cells of sponges and other metazoans. However, the evolutionary origin of this homology is debated. While some evidence suggests the collars of choanoflagellates and sponges arose independently by convergent evolution (Mah et al, 2014; Sogabe et al, 2019), other studies provide counter-evidence for a shared origin based on the deep conservation of the underlying molecular toolkit (Peña et al, 2016; Brunet and King, 2017). Thus, the direct evolutionary lineage between these two cell types has been called into question.

The synzoospore model added an important concept: that the unicellular ancestor of animals was an organism with a complex life cycle capable of transitioning between different life stages depending on environmental conditions, similar to what is observed in filastereans and ichthyosporeans (Mikhailov et al, 2009). However, this model retains from the choanoblastea model the assumed stepwise transition from choanoflagellate to a sponge, with the flaws outlined above. A helpful addition to the synzoospore model was the idea that the unicellular ancestor had different cell types that could undergo temporal transitions, and that a major innovation was the ability to maintain different cell types simultaneously, perhaps in a spatially organized manner. In thinking about genetic innovations, both of these models implied that there was extensive genetic innovation arising once they started becoming animals to produce many biological innovations (such as complex cell signaling, spatially differentiated cell types, cell–cell and cell–extracellular matrix adhesion, and complex transcription factor regulatory networks). However, as noted earlier, much of this innovation is already present in unicellular protists related to animals.

In summary, what the fossil record told us (an explosive origin of animals), the prevailing understanding from model organism genomes (that animals are much more complex than unicellular eukaryotes), and the two major models of animal origins (choanoblastea and synzoospore) have collectively painted a picture of animal evolution as a progression from simple to more complex organisms closely resembling living organisms and requiring extensive genetic innovation. However, this picture needs to be revised, as the discovery of additional unicellular organisms closely related to animals (Hehenberger et al, 2017; Tikhonenkov et al, 2020; Urrutia et al, 2021), along with revelations of their genomic and cellular sophistication (Ros-Rocher et al, 2021), show that the genetic machinery required to evolve animals had been steadily increasing in unicellular organisms, requiring much less of a jump to form the first animal than previously envisioned.

## Key ancestors in the path towards animals: LUAA and FACA

To address the origin of animals, we need to consider three crucial extinct ancestral organisms: 1) the Last Unicellular Ancestor of

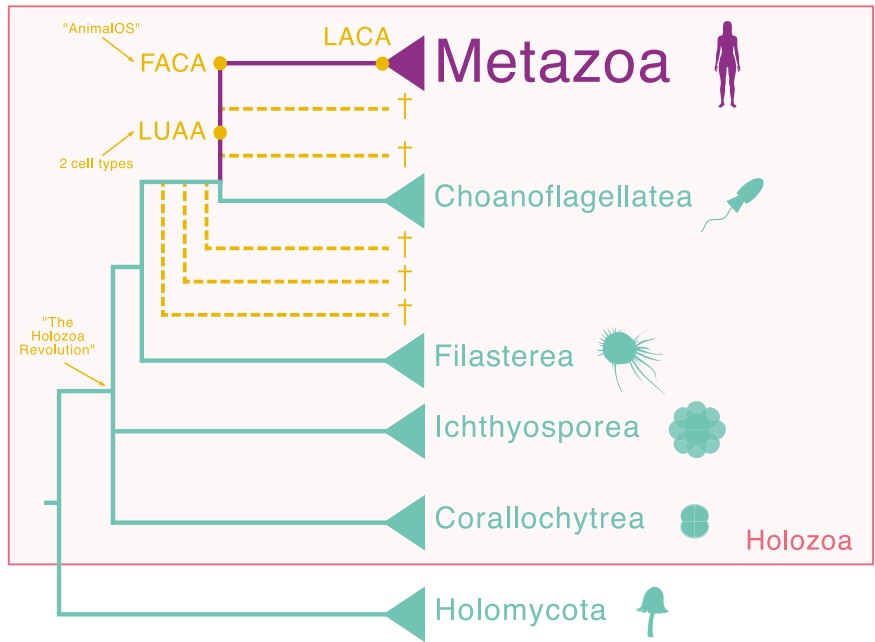

**Figure 2.  A schematic tree of Opisthokonta, illustrating its different ancestors and key to understanding the transition.**

LUAA Last Unicellular Animal Ancestor, FACA First Animal Common Ancestor, LACA Last Animal Common Ancestor. Relationships based on Torruella et al, 2012; López-Escardó et al, 2019; Grau-Bové et al, 2017; Ocaña-Pallarès et al, 2022.

Animals (LUAA), 2) the First Animal Common Ancestor (FACA), and 3) the Last Animal Common Ancestor (LACA) (Fig. 2). LUAA represents the unicellular organism whose innovation paved the way for animal evolution. A subsequent period of innovation led to FACA, the first true animal, which we will define later. LACA is the ancestor of all living animals, and it is possible that FACA and LACA are the same, or that a significant period of further innovation occurred between them. Therefore, to address the origin of animals, we need to generate a better picture of LUAA and FACA. Deciphering the nature of these ancestors requires rigorous comparative analyses between extant animals and their closest living unicellular relatives.

It is also important to situate LUAA in the tree of eukaryotes. We know that several branches of eukaryotes rapidly diverged from the last eukaryotic common ancestor (LECA), including those leading to plants and animals (Fig. 1) (Simpson and Roger, 2004; Burki, 2014). Here, we will analyze the whole evolutionary trajectory from LECA to FACA, but will focus specifically on the eukaryotic branch Amorphea, which eventually gave rise to the Opisthokonta group (Fig. 1), encompassing animals and their closest unicellular relatives (the Holozoa), as well as fungi and their closest relatives (the Holomycota) (Ruiz-Trillo et al, 2008; Torruella et al, 2012).

Interestingly, this evolutionary trajectory from unicellular organisms to animals shares parallels with the origin of eukaryotic cells from prokaryotic ancestors. In that scenario, a last archaeal common ancestor transitioned into the first eukaryotic common ancestor, followed by a long period of gene innovation to generate LECA (O'Malley et al, 2019; Roger et al, 2017; Eme et al, 2017; Vosseberg et al, 2024). Crucially, somewhere between the first and last ancestors, an archaeal cell fused with a bacterial cell—a key symbiotic event that led to the acquisition of mitochondria. Similar

to animal evolution, the sequencing of new Archaea species has already reduced the number of proteins previously thought to be unique to eukaryotes. Thus, insights from broader sequencing of extant species can continue to inform our understanding of how the first animals evolved.

Our focus here will be on identifying the essential elements that gradually evolved along the protist-to-animal trajectory. However, data alone are not enough to retheorize how animals evolved. We need a more effective way to frame this series of innovations and we see analogies with technological evolution, in particular the evolution of the smartphone, that we believe are very productive to explore.

## Biology meets technology: the origin of animals vs the origin of the smartphone

Technological evolution offers a valuable lens through which to examine biological innovation (Basalla, 1988; Tëmkin and Eldredge, 2007; Arthur, 2009; Solé et al, 2013). Indeed, the unfolding of technology reveals contingencies, convergences, tinkering, stagnation, and major transitions that often parallel some of their living counterparts (Solé et al, 2011). Moreover, technological and economic perspectives of innovation can help us understand universal requirements for the emergence of novelty and its role in niche construction (Erwin, 2008). This leads to an intriguing question: are there technological analogs that can effectively illustrate the complex process of the origin of animals? As mentioned above, we propose that the evolution of the smartphone bears striking similarities to the evolutionary journey that led to the emergence of animals.

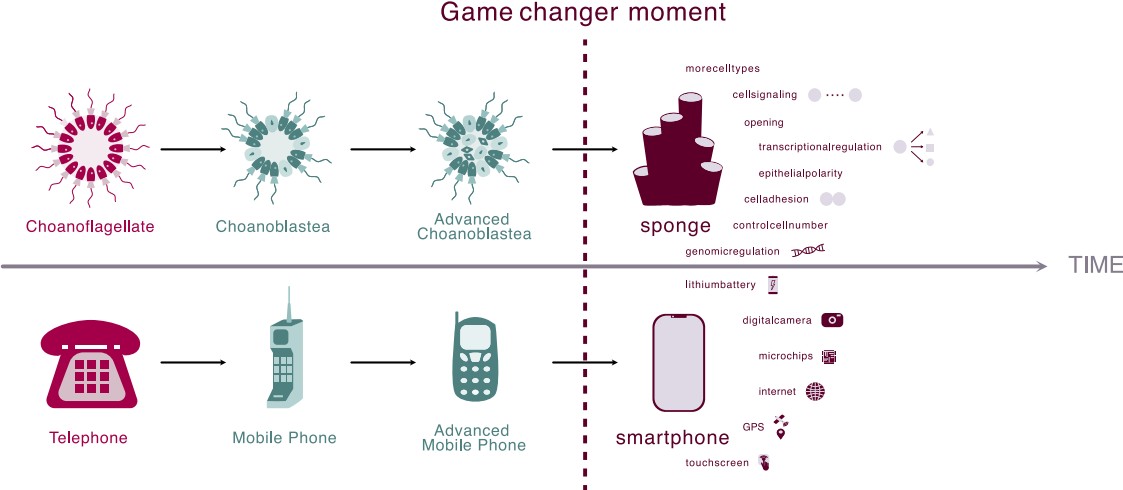

**Figure 3. A "choanoblastea" vision on how smartphones evolved.**

Applying the ideas of the choanoblastea and synzoospore models to the origin of the smartphone highlights the limits of these two models. A "choanoblastea" vision of the smartphone evolution would suggest a direct progression from landline phones to early, short-range mobile phones, and then suddenly to the modern smartphone with all its features, such as internet connectivity, GPS, camera, and so on; suggesting all these advanced technologies—MP3 players, internet access, gaming platforms, and miniature camera sensors—emerged simultaneously at the onset of the first smartphone. However, this is not how it occurred. Instead, the origin of the smartphones was the result of the gradual convergence of technologies independently developed by various industries, including computing, photography, gaming, and music (see Fig. 4).

To start with, this analogy illustrates the limitations of the choanoblastea and synzoospore models. Applying the "explosive" origin and linear or stepwise evolution concepts (inherent in these two models) to the origin of the smartphone would suggest a direct progression from landline phones to early, short-range mobile phones, and then suddenly to the modern smartphone with its internet connectivity, audio capacities, GPS, camera, and other advanced features (Fig. 3). This implies that all these constituent technologies—MP3 players, internet access, gaming platforms, and miniature camera sensors—emerged simultaneously, resulting in the first smartphone. However, this is not how it occurred. Instead, the origin of the smartphones was the result of the gradual convergence of technologies independently developed by various industries, including computing, photography, gaming, and music (Fig. 4). These technologies matured over time until they were integrated into a single device. This integration was driven by sophisticated software, the operating system (OS), that allowed the evolution of smartphones. Indeed, the evolution of both animals and smartphones relied on the maturation of numerous prior innovations until they became compatible for integration into either a portable format or a multicellular entity.

Thus, we envision the origin of both smartphones and animals as a prolonged evolutionary process, divisible into three distinct waves of innovation. First, a "foundational era" when the hardware of electronics and eukaryotic cell function evolved. By hardware, we mean the basic technological and biological functions. This was followed by a "compounding era", marked by an acceleration of innovations that pre-equipped devices and cells to eventually become a smartphone and animals, respectively. Finally, a "game-changer" moment occurred, where smartphones and animals emerged, leading to the creation of novel markets and biological ecosystems, respectively. We next elaborate on each of these waves (see Fig. 4).

## The foundational era: evolution of innovative yet basic functions necessary for the origin of smartphones and animals

The progression of both technology and life hinges on foundational prerequisites that enable novelty to occur (Burke, 2007; Kauffman, 2000). In the case of the smartphone, the rise of modern electronics required first the development of a toolkit of components (resistors, capacitors, coils, batteries and so on), culminating in the crucial breakthrough of the transistor. This established the necessary toolkit for subsequent advancements.

Similarly, the evolutionary trajectory toward animals relied on ancestral biological capabilities. The prior evolution of the first eukaryotes, with its energy-generating mitochondria and compartmentalization, is essential for further complexity. Moreover, most of the cell morphologies and behaviors that we see today in extant animals were already present in more ancient eukaryotic ancestors (flagellates, amoebas with filopodia, aggregation, chemotaxis, etc.), including organisms within Amorphea and the CRuMs group (see Figs. 1 and 4). Furthermore, these ancestors possessed sophisticated control over and a remarkable ability to transition between different cell stages, primarily triggered by fluctuating environmental conditions. At a genome level, those ancestors already had genes key for animal multicellularity, including genes whose products are involved in cell adhesion, cell signaling and transcriptional regulation of pattern. Even members of the CRuMs group possess genes initially thought to be animal-specific due to their absence in yeasts, such as intracellular components of the integrin adhesion machinery (Kang et al, 2021). It is worth noting that pan-eukaryotic analyses have also provided solid data about the wider presence of certain gene families, such as *septin*, *integrins*, or specific protein modifiers, in different eukaryotic lineages informing us of what was already available in the Last Eukaryotic

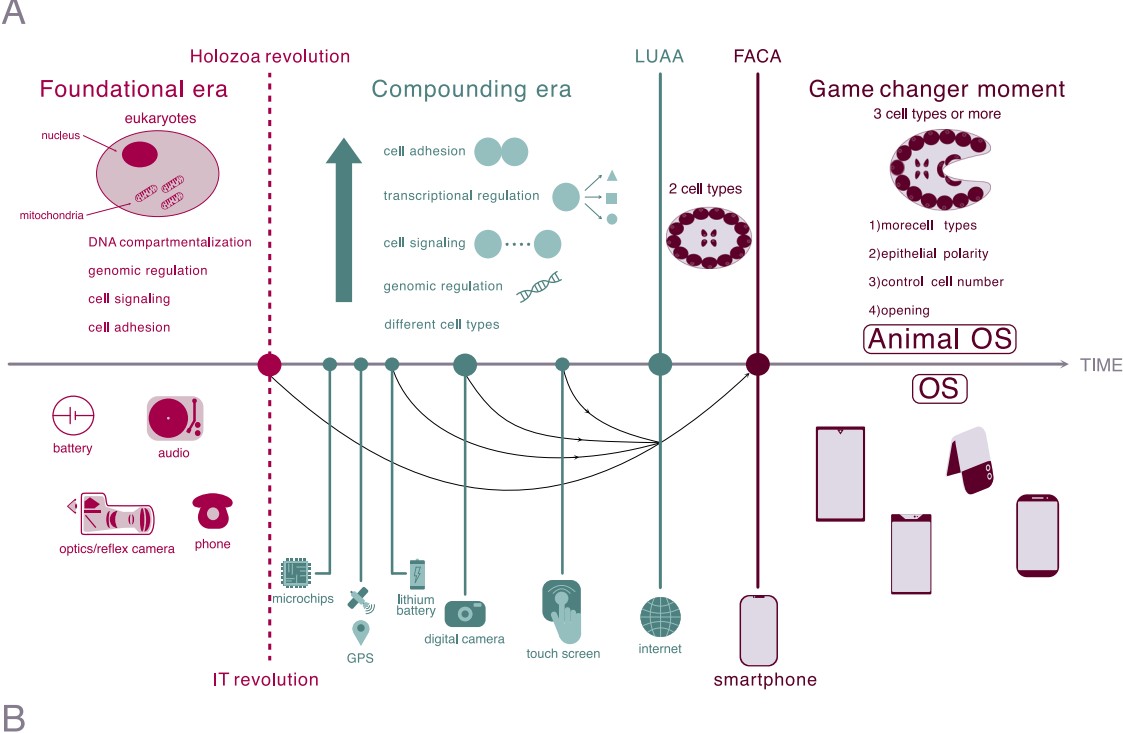

**Figure 4.  A comparative timeline of animal origins and smartphone emergence.**

This dual-scale timeline highlights the parallels between biological (**A**) and technological (**B**) evolution. In both cases, we delineate the three eras: (1) a "foundational era"; (2) a "compounding era", and (3) a "game-changer" moment enabled by the establishment of "OS" (**B**) and "Animal OS" (**A**) systems.

Common Ancestor (LECA) (Delic et al, 2024; Kang et al, 2021; Weiner et al, 2020). These ancestral biological "technologies" were analogous to the hardware components (sensors, processors) and basic software development kits (SDKs) that existed before the advent of smartphones. These pre-existing elements were adapted and play a crucial role in complex animals and the modern smartphone. The development in both domains is thus characterized by a building-block approach, where earlier innovations and features provide the groundwork for subsequent, more complex forms (Fig. 4).

## The information technology and the Holozoan revolution

A turning point in the technological realm was the rise of information technology (IT) in the 1950s, as the world transitioned to a digital era (Augarten, 1984; Valverde, 2016; Campbell-Kelly et al, 2023). Early landmarks included phones (and telephone networks), early audio tech devices and optical cameras. The rise of the IT era (thanks to the invention of the transistor, which allowed the building of integrated circuits) marks a rapid progression in inventions related to digital systems, new materials, and networking (Fig. 4). They provided the ecological context (GPS, Internet, shared computer software) and the technological toolkit that made the smartphone possible. This toolkit included crucial elements such as microchips (enabling integration, information storage, and processing, including music and video), miniaturized digital

cameras (rapidly shrinking in size), digital audio (MP3), new batteries (small and long-lived), and screens based on liquid crystal displays. Each of these parts was in place before the first smartphones emerged, but initially, they were employed in less powerful gadgets. Therefore, before the advent of the first true smartphones, a landscape of independently developed technologies and less-integrated devices paved the way for their eventual convergence into the smartphone. Within the technological context, where the long-term success of a given innovation relates to its "propagation" among potential users, integration would be favored by versatile, small-sized devices.

In the biological context, the turning point was the origin of Holozoa (Fig. 4). While the Amorphea branch of eukaryotes had already innovated new proteins (many of which were subsequently lost in the fungal lineage), the origin of Holozoa saw an acceleration in the rate of acquisition of novel proteins shared with animals (Ocaña-Pallarès et al, 2022). What were the biological technologies that created this Holozoa revolution? We and others have already reviewed how several groups of unicellular organisms closely related to animals share a surprisingly large number of genes previously thought to be key innovations of animals, and are unique to this branch of the eukaryotes (reviewed in Sebé-Pedrós et al, 2017; Ros-Rocher et al, 2021; Ruiz-Trillo et al, 2023). We will here just have a short summary of these. The pre-metazoans holozoans already had the following machineries: (1) cell adhesion, with many components of integrin-mediated adhesion and the beginnings of cadherin and lectin mediated cell–cell adhesion (Sebé-Pedrós et al, 2010; Nichols et al, 2012; Suga et al, 2013); (2)

signal transduction, including the Hippo pathway, tyrosine kinases and an early precursor of the Notch pathway (Suga et al, 2013; Suga et al, 2012; Sebé-Pedrós et al, 2012; Phillips and Pan, 2024); (3) novel types of transcription factors involved in animal development such as LIM Homeobox, nuclear factor-κB, p53/63/73, RUNX, and T-box (including Brachyury), and those specifying stem cell lineages, such as Sox and Pou (Sebé-Pedrós et al, 2011; Suga et al, 2013; Grau-Bové et al, 2017; Gao et al, 2024); and (4) additional components of more complex gene regulation, such as cell-type specific control of chromatin accessibility, cell-type specific phosphorylation of TFs, cell-type specific alternative splicing (including exon-shuffling), cell-type specific long-non-coding RNAs, and animal-type miRNAs and the associated microprocessor complex (Sebé-Pedrós et al, 2016; Grau-Bové et al, 2017; Grau-Bové et al, 2018; Brate et al, 2018). These organisms also acquired the ability to form different cell types with diverse morphologies, such as amoeboid or more spherical; they have protrusions, including filopodia, stalks, or flagella; and a protective cell wall may surround them (Ros-Rocher et al, 2021). These protists can also form diverse kinds of multicellular structures, which include those formed by clonal division in choanoflagellates, ichthyosporeans, and corallochytreans, as well as aggregates of thousands of cells in filastereans (Ros-Rocher et al, 2021; Li et al, 2025; Bercedo-Saborido et al, 2025; Ros-Rocher et al, 2023) and a few choanoflagellates (Ros-Rocher et al, 2025). The first examples of a multicellular structure containing morphologically distinct cell types were recently described (Ruiz-Trillo et al, 2023; Shabardina et al, 2024; Laundon et al, 2019; Olivetta et al, 2024). Still, to date, there is no clear example of a stable multicellular structure with reproducible spatial organization of different cell types in Holozoa.

Thus, our improved understanding of the genome content and cellular biology of unicellular organisms closely related to animals allows us to infer that the ancestors of Metazoa had much of the cellular machinery involved in animal development and home-ostasis. Notably, there was an acceleration of innovation of both novel protein domains and novel proteins that created a "compounding effect": novel protein domains facilitated the emergence of even more novel genes within the same functional categories. Therefore, this "compounding effect" resulted in a "snowballing" effect that culminated in LUAA (Fig. 4). It is worth noting that these individual biological components may not have initially evolved with multicellularity as their primary function, just as early technological innovations were made before the idea of a smartphone arose. Rather, the biological 'ecosystem' selected for the accumulation, interaction, and refinement of these innovations, which eventually paved the way for the emergence of the animal kingdom. In contrast, for example, in the fungal lineage, a different path of gene innovation and loss resulted in a different kind of multicellular organism than animals (Ocaña-Pallarès et al, 2022). Within our technological analogy, the start of the Holozoan Revolution would be the counterpart of the invention of the transistor and the subsequent, rapid expansion of integrated circuit designs.

How did LUAA evolve from this Holozoa revolution? We propose an important step was the conversion from temporal cell types (or temporary spatially differentiated cell types without clear division of labor) to simultaneous cell types organized in a multicellular structure. Before LUAA, we had unicellular organisms with signaling pathways that responded to environmental signals by changing the cell's fate, both morphologically and metabolically. They have two or more potential cell types that are adopted depending on the environmental conditions, and, in some specific moments, those cell types may co-exist as we see on some living non-animal holozoans. From this, an organism evolved with two stable, co-existing cell types maybe through the innovation of surface ligands that mimicked environmental signals, allowing each cell to maintain the fate of the other. This represents the innovation in LUAA. These two cell types became reliant on each other for survival due to complementary loss of function and specialization. In other words, the two cell types split their activities. This reciprocal change in activity would maintain both cells, as each became dependent on the other and lost the ability to survive independently. This has similarities to the symbiotic fusion event between archaea and bacteria, thought to have generated the eukaryotic cell. However, in this case, it is a "self-symbiosis" between two cell states that previously operated independently. While unicellular organisms can transition between different cell stages, each stage must be capable of surviving in a changing environment. We envision a substantial advantage in one cell being able to reduce its expression of the genes required for activities that the other cell type is undertaking, as this will enable the cell to become more specialized and permit innovation in cell type capabilities. Similar to technological evolution, enhancement in one cell type could also drive enhancement in the other, creating a positive feedback loop.

This separation of two cell types likely involves, among other things, modifications to the gene regulatory networks, which are split, so that the new cell signals lead to a transcription program that specifies cell morphology and differentiated functions. In contrast, environmental signals still regulate the genetic program that modifies metabolism. A second key step is to control the spatial organization of these two cell types reproducibly. This can be achieved by modest changes to the existing cell adhesion machinery, as the cell adhesion components were already complex; for example, C. owczarzaki has four integrins and one cadherin (Suga et al, 2013). A simple duplication of cadherin and differentially regulated cadherins and integrins between cell types could easily create two cell layers, each stabilized by the cadherins, and connected via the extracellular matrix, as occurs in animals. Thus, a loose aggregate of cells would easily be transformed into an organized structure, much like the early steps in animal embryogenesis. In its simplest form, the multicellular stage of this organism would have been a hollow ball or cup with an outer layer of cells specialized for protection from the environment (at the cost of reduced food intake), while the inner layer specialized for food digestion and nutrient transfer to the outer layer, perhaps with the aid of a symbiotic microbiome, at the expense of losing protective functions.

We envision that these early precursors of animals would still transition between multicellular and single-cell forms. After all, every animal still starts life as a single cell, even if that cell does not normally exist as a free-living unicellular form capable of reproduction by division. It is worth noting that in that case, cell signaling had to be maintained to respond both to environmental conditions and to maintain the integrity of the multicellular entity. Moreover, cell plasticity had to be high enough to respond to different environmental cues and survive changing conditions,

while retaining this plasticity in a dormant state during the multicellular life stage. What gave the multicellular form of this organism a selective advantage? We suggest that the establishment of two cell types that coordinate their activities, and have become specialized, is also the beginning of homeostasis in the form of the ability to stabilize the environment surrounding the cells. As the outer layer of cells became more impermeable, and any opening could be closed and opened, the environment for the inner layer of cells would be under the organism's control. This would permit the inner cells to further reduce the energy devoted to being able to respond to environmental change, and amplify and enhance novel properties of the cellular repertoire at their disposal. An alternative explanation is that these pre-metazoan organisms were slowly losing their ability to fastly adapt to changing environments (perhaps too costly to maintain all those signaling machines), and, somehow, they became forced to keep the multicellular stage for longer. If the multicellular stage persists long enough, the cells must adapt to this multicellular microenvironment, which includes biomechanical constraints different from those encountered in the unicellular stage. Under this condition, changes in the cytoplasm or cytoskeleton organization or even in the production of specific proteins (such as secretion or the positioning of specific adhesion molecules to the surface) could provide a specific phenotypic signature, initially environmentally induced, that could be reinforced due to prolonged exposure to this multicellular stage, inherited cortically, and even become fixed by epigenetic assimilation, eventually canalizing the evolution to a fixed multicellular entity (Sonneborn, 1970; Muroyama et al, 2023; Ashe et al, 2021; Weiner and Katz, 2021).

Even though LUAA was not yet an animal, it was already preadapted to become an animal. LUAA was like those early devices combining phone functions with a basic personal digital assistant, or a camera with a phone—functional, but not yet a game-changer. It is worth noting that in our view, and in contrast to other views (see, for example, Mikhailov et al, 2009) the temporal (or temporary spatial) to spatial cell differentiation occurred before the origin of animals.

## Smartphone OS and "Animal OS": the crucial role of integrative software

The final, critical step in the transition from unicellular ancestors to animals, or from pre-smartphone devices to smartphones, was the seamless integration of pre-existing components into a cohesive, functional entity. In the technological realm, small-sized components could work interdependently thanks to software running on small circuits. Software and hardware were tightly integrated, and over time, improvements in one component, such as the camera, drove enhancements in others, including storage capacity and processing power, thereby making the whole system more cohesive and optimized for mobile use. A key breakthrough that enabled the emergence of the first smartphones was the creation of a sophisticated, extensible operating system—one capable of seamlessly integrating various technologies in a way that had not been possible before. This integration generated a powerful network effect.

We hypothesize that the evolutionary transition from LUAA to FACA may have followed a similar principle: rather than requiring many new components, it likely involved the development of an "Animal Operating System". We define this Animal "OS" not as a single component, but as the hierarchical integration of multiple pre-existing systems into a cohesive developmental program. This includes the core Gene Regulatory Networks (GRNs) that define cell identity (the "kernel"), the cell–cell signaling and adhesion pathways that manage inputs and outputs, and the mechanisms that control cell number and stoichiometry (the "resource management"). The later evolution of more complex gene regulatory networks and chromatin architecture in bilaterians can be seen as a major "OS upgrade", enabling more sophisticated body plans. It is worth noting that, despite their substrate differences, there are remarkable similarities between the network organization of gene regulatory networks and the architecture of operating systems (Yan et al, 2010).

To evolve from LUAA to FACA, we hypothesize that four additional innovations were required, which also mirror some of the innovations of smartphones. First, the further diversification of cell types eventually led to the formation of the first organs. Among these were coordinated proto-muscles, which enabled a new scale of rapid, multicellular contraction that added a more powerful way of locomotion to the metazoan repertoire. This allowed the organisms to outsource the creation of key amino acids, for example (see below). We envision that contraction was first stimulated by inherent mechanosensitive pathways, eventually evolving into specialized sensory cells, the precursors to the nervous system. This is akin to smartphones developing more specialized hardware and a rich app ecosystem, moving beyond basic functions. Second, the development of apical–basal epithelial polarity was crucial in consolidating the inside-out differences, thereby providing a homeostatic environment for the internal cells. This mirrors the development of a nice, user-friendly screen, shielding together with the case—the complex internal hardware and software—creating a clear "inside-out" differentiation. Third, the development of mechanisms to control cell number and the stoichiometry of cell types. This is analogous to sophisticated OS resource management (CPU, memory, battery), ensuring all components work cohesively and efficiently. Fourth, the establishment of pattern formation, including the formation of at least one stable opening (mouth/anus) to permit ingestion of large particles of food. Similarly, smartphones have finally evolved standard charging/data ports. Once those four innovations were acquired, this would qualify as FACA, and the diversification of animals could begin.

This raises the question of how we define an animal. If we discover an organism that branched off the tree between choanoflagellates and sponges/ctenophores (Figs. 1 and 2) and is multicellular, should we call it a protist or an animal? We hope to face this dilemma soon, as we have recently discovered, through environmental sequencing of ribosomal DNAs, that there are at least eight lineages of unicellular relatives of animals that have yet to be observed (Ruiz-Trillo et al, 2023; Ruiz-Trillo, unpublished data). Previous definitions of Metazoa were based on those animals already identified (e.g., Adl et al, 2012), and therefore do not provide a guideline for a simpler version. The four innovations that we envision are needed to become FACA, which provide a set of criteria to define whether an entity is an animal or not. Therefore, we propose that an animal will be a multicellular holozoan, with at least three cell types spatially arranged in a reproducible way. The outer layer of this animal exhibits apical–basal polarity, features an

opening for food ingestion, and possesses the capacity to contract or swim away if poked. This can also be paralleled by the smartphone, which is a portable (motility), integrated (multicellular) computing device with multiple Apps (body plans, ecological strategies), input/output modalities (an opening), a case (the epithelium), and complex responsiveness (contractile capacity).

## Additional similarities between animal evolution and the smartphone

Both animals and smartphones triggered a significant transformation and expansion of their respective ecospheres. Smartphones became a central piece for "ecosystem engineering" (Erwin, 2008; Standage, 2021) as competing devices faded. They accelerated the development of new niches and technological diversity, leading to an explosion of mobile apps that reshaped everything and drove advancements in wireless networks, cloud computing, and data storage. Indeed, each new app increased the platform's value, attracting more users and developers in a virtuous cycle. In the case of animals, once the "Animal OS" became robust and versatile enough, it could enable the rapid diversification of "applications" (such as the formation of the App Store)—including new body plans, ecological strategies, and functionalities. Therefore, the invention of "Animal OS" could lead to a rapid deployment of pre-built potential, which could explain the rapid diversification seen in the fossil record. The period before the Cambrian explosion, where animal ancestors might have been small and too simple to fossilize, could be analogous to a period of "beta testing," resulting in the development and refinement of the Animal OS.

The process of co-option, where a protein evolved for one function is adapted to a new function, also applies to both the origin of animals and the origin of the smartphone. In the origin of animals, we suspect that the original use of proteins in the unicellular ancestor was often adapted to new roles within a multicellular entity. For example, some transcription factors may have had a role in the transition between different life stages in premetazoan organisms, being later on co-opted to regulate different cell types. Or genes that may have had a role in interpreting the environment could have cbeen o-opted to sense signals from other tissues from the same organism. Similarly, some components of the smartphone are now being repurposed for new functions, such as the sensors originally used to determine orientation, which can now be utilized for health monitoring or other functions. This integration of pre-existing biological or technological modules into a multicellular organism (FACA) or a smartphone reshaped life's and market's ecosystems, showing rapid diversification.

The smartphone analogy can be extended to another aspect of animal evolution: secondary simplifications. This refers to the process by which the adaptive pressure of specific environments may lead to a simplification of the genome or the morphology and the functional properties it produces (O'Malley et al, 2016). Our analogy highlights at least two distinct modes of this process.

The first is functional outsourcing, where an organism sheds a costly internal function that can be reliably obtained from its environment (Domazet-Lošo et al, 2024). The loss of metabolic genes at the root of the animal tree, including the inability to synthesize nine essential amino acids, is a very good example

(Richter et al, 2018; Ocaña-Pallarès et al, 2022; Domazet-Lošo et al, 2024). This metabolic simplification was likely driven by the evolution of more efficient feeding, which outsourced this biochemical requirement to external food sources. This is strikingly parallel to modern smartphones, which have outsourced vast amounts of memory and computational power to the "cloud", allowing the devices themselves to become leaner while accessing more power than they could ever hold internally.

The second mode is niche specialization, where features are shed to optimize a device or organism for a specific role (O'Malley et al, 2016). For example, there are now smartphones that feature large screens and extensive visual displays but fewer of the technologies present in most smartphones. They occupy a specialized technological/ecological niche for users who only need the phone component with a large display. Another example could be the iPod shuffle, which offers music listening in a minimal space and weight by stripping away most smartphone features (Fig. 5). Importantly, those secondary simplifications are not a copy or a reversion to the precursor mobile phone or portable audio cassettes. They retain some of the innovations that arose in the smartphone, such as the use of WiFi or the "OS". In a similar way, animals like sponges or placozoans might represent a "low-spec" version of multicellularity, retaining core features but shedding complexity for efficiency in certain, specific environments and lifestyles. Sponges, for example, may or may not have shed complex tissues and organs to become highly efficient at filter-feeding, thereby solving energy bottlenecks in specific environments (Asadzadeh et al, 2020). Nevertheless, the alternative view that sponges instead may represent the ancestral body plan of animals remains an unlikely but valid possibility.

## Conclusions

How do innovations occur? This is an old question that has been explored by scholars across various disciplines, including evolutionary biology and technology (Tëmkin and Eldredge, 2007). Although there is no universal definition, innovations are typically understood as novel traits or features that emerge in a lineage and enable new functions or ecological opportunities, potentially leading to increased diversification or adaptive success (Wagner and Lynch, 2010; Erwin, 2015). The history of life on our planet is marked by such innovation events, including the so-called major evolutionary transitions (Maynard Smith and Szathmary, 1995; Szathmáry, 2015; Lane, 2010), from the origins of life, cells, and multicellular life to language and cognition. Very often, the same innovations occur independently multiple times with similar outcomes, suggesting that convergent evolution is a dominant process (Conway Morris, 2003; Powell, 2012; Solé et al, 2024). Each innovation opened up new niches by giving organisms novel traits that allowed them to exploit previously inaccessible environments or resources. For instance, wings allowed insects to access the air, creating new ecological roles and opportunities, while lungs enabled vertebrates to move onto land. But some particular innovations profoundly changed the biosphere and can be considered Black Swan events: they happened only once, and a high impact accompanies their rarity (due to the need for many factors required) (Ruiz-Trillo et al, 2023). One such event was the rise of animals, which might have been a unique occurrence, both

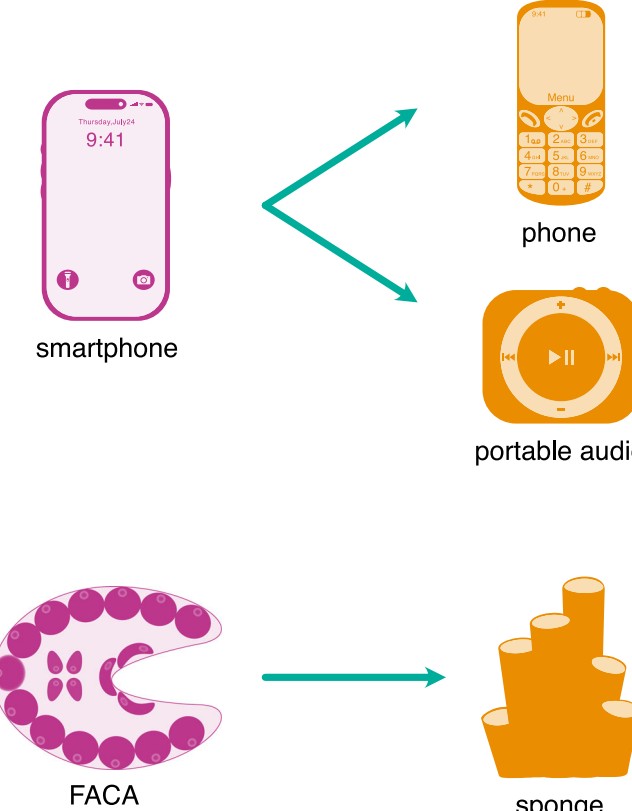

**Figure 5. Examples of secondary simplifications in both technology and biology.**

A complex smartphone (left) gives rise to functionally specialized devices (right) that strip away non-essential features (e.g., cameras, apps) while retaining core innovations, such as modern operating systems and wireless technology. Similarly, sponges (right) exhibit a simplified body plan compared to other animals or FACA (left). It has been proposed that living sponges lack true tissues and organs, a successful adaptation for a filter-feeding lifestyle, yet they are built from the same fundamental molecular toolkit for multicellularity as more complex animal relatives.

in terms of its requirements and its macroevolutionary consequences (Erwin, 2015). What was special about it?

As we have discussed, the emergence of the first animals required preconditions already present in their unicellular ancestors, including genes for cell adhesion, signaling, and differentiation. These molecular toolkits were co-opted, allowing for the coordination, specialization, and structural organization necessary for animal life to evolve. Although the emergence of animals was a rare event, we have explored a parallel process of technological evolution that, in many ways, has taken place through a similar process, despite the known fact that biological systems evolve through tinkering (Jacob, 1977; Solé et al, 2002). The smartphone analogy is more than a metaphor here, since it shares multiple commonalities despite the apparent differences, and it provides food-for-though in the neutral analogy realm. It required the previous origination and subsequent change (by design, not through Darwinian evolution) of multiple components (including hardware and software), each with relevant functionalities that were useful in specific contexts. While the ecological context for

eukaryotic cells grew in a world dominated by microbial mats, technological designs "propagate" within populations of users. In both cases, the precursor components expanded in their respective landscapes as they favored diverse functional traits. The invention of the smartphone became possible through the combination of context (communication networks), market needs (multi-use devices), and efficient, scalable technology (compact components). Once these elements converged within a single system, they not only succeeded as a unique innovation but also created a whole new technological, social, and economic landscape of opportunities. These new opportunities, in turn, favored expansion and diversification.

## Future directions

How might the smartphone analogy empower future research on the question of the origin of animals? First, it provides some concrete and testable predictions about the order of events between LUAA and FACA. For example, our analogy, which distinguishes between the acquisition of "hardware" (genes) and the subsequent development of a unifying "operating system" (our FACA criteria), leads to a concrete and testable prediction. Specifically, we predict the existence of extant or extinct lineages that represent these intermediate "prototypes"—organisms that have achieved stable, spatially organized multicellularity with interdependent cell types (our proposed LUAA stage), but which lack the full "OS" of true animals, such as consolidated epithelial polarity or a defined opening (our proposed FACA stage). The ongoing search in one of the author's lab for new, deep-branching holozoan lineages provides a direct avenue to test this prediction (del Campo and Ruiz-Trillo, 2013; Ruiz-Trillo et al, 2023). This analogy also highlights how the journey from single cells to complex animals, similarly to the evolution of smartphones, was fueled by the power of integration and repurposing of pre-existing modules. Inspired by this engineering perspective, experimentalists can attempt to reconstruct ancestral states in living unicellular relatives—such as *Capsaspora*—by targeted modification. For example, tweaking cell-adhesion molecules and signaling pathways, then coupling them to a minimal "integrative software" implemented as a synthetic gene-regulatory network, could stabilize a multicellular stage with more than one cell type. Beyond experimental inspiration, we see the smartphone analogy as providing a fresh and productive conceptual framework for understanding this major evolutionary transition, challenging previous preconceptions and inviting us to examine more analogies between technology, society, and evolution.

## Peer review information

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

## Acknowledgements

We thank Maureen O'Malley and Joaquín Gabas for advice and discussions, and Meritxell Antó for advice and help on the figures. Work at the IR-T and EC lab is co-funded by *Grant PID2023-153273NB-I00 funded by MICIU/AEI/ https://doi.org/10.13039/501100011033 and "ERDF/EU", by the "European Union", as well as by the European Union (ERC, MISSINGRELATIVES, 101097659). Views and opinions expressed are, however, those of the author(s) only and do not necessarily reflect those of the European Union or the European Research Council. Neither the European Union nor the granting authority can be held responsible for them. We also acknowledge support to the Departament de Recerca i Universitats de la Generalitat de Catalunya (exp. 2021 SGR 00751). RS acknowledges the support of the Santa Fe Institute.

## Author contributions

**Iñaki Ruiz-Trillo**: Conceptualization; Writing—original draft; Writing—review and editing. **Elena Casacuberta**: Conceptualization; Writing—original draft; Writing—review and editing. **Nicholas H Brown**: Conceptualization; Writing—original draft; Writing—review and editing. **Ricard Solé**: Conceptualization; Writing—original draft; Writing—review and editing.

## Disclosure and competing interests statement

The authors declare no competing interests.

