## [Peer Review File · The EMBO Journal]

A smartphone analogy to explore the origin of animals

Iñaki Ruiz-Trillo, Elena Casacuberta, Nicholas Brown, and Ricard Solé

Corresponding author: Iñaki Ruiz-Trillo (inaki.ruiz@ibe.upf-csic.es)

Review Timeline:

Submission Date:	1st Aug 25
Editorial Decision:	14th Sep 25
Revision Received:	20th Oct 25
Editorial Decision:	11th Nov 25
Revision Received:	11th Dec 25
Accepted:	17th Dec 25

Editor: Yehu Moran

Transaction Report:

Dear Prof. Ruiz-Trillo,

Thank you for submitting your interesting review manuscript to the EMBO Journal. It has now been seen by four expert referees whose comments are enclosed. As you will see, all referees express interest in your manuscript and are broadly in favour of publication, pending satisfactory minor revision.

Given the referees' positive recommendations, I would like to invite you to submit a revised version of the manuscript, addressing the comments of all reviewers.

Thank you for the opportunity to consider your work for publication. I look forward to your revision.

Yours sincerely,

Yehu Moran
Editor
The EMBO Journal

Referee #1:

The current manuscript presents a origin of multicellular animals as a technological analogy. The review is well written and will likely be of interest to many readers of the EMBO journal. While I wholeheartedly agree that the smartphone analogy is very useful in conceptualizing the transition between coloniality and multicellularity, the manuscript contains views on sponge biology that are not necessarily correct - and importantly, are not key to the analogy used by the authors. Specifically:

"Although sponges may possess the simplest multicellular structure among extant animals"

- This is not true, Placozoans are much simpler, both in terms of overall morphology, size, and number of cell types. Consider

giant barrel sponges or deep water carnivorous sponges and compare these to Trichoplax....

"Third, the choanoblastea model was in part based on the perceived homology between an unusual cellular structure called the collar complex"

- The collar complex is not an unusual structure; in fact collar cells are present in all animals except Ctenophores and majority of Ecdysozoans - see Brunet and King 2017

"However, recent morphological and molecular evidence suggests they arose independently by convergent evolution (Mah et al. 2014; Sogabe et al. 2019)."

- This is one viewpoint, but the issue is far from settled. See Peña et al 2016 for an alternative view, which should be considered and cited alongside those referred to by the authors.

"First, the further diversification of cell types eventually led to the formation of the first organs, including proto-muscles that could generate rapid, multicellular contractions, thereby maintaining motility as a key aspect of metazoan multicellularity."

- Many animals, especially aquatic larvae - but also adult flatworms and ctenophores - use cilia rather than muscle for movement. Given cilia were almost certainly present in the ancestors of animals, the first animals did not have to evolve muscle for movement.

"In a similar way sponges might represent a "low-spec" version of multicellularity, retaining core features but shedding complexity for efficiency in certain, specific environments and lifestyles."

- The authors' view that sponges are secondarily simplified is as valid as the alternative view - that their body plans represent ancestral condition for animals. Again, Trichoplax would be a "safer" example here.

"Moreover, simplification often solves energy bottlenecks by focusing, for example, on more efficient consumption, something that sponges excel at."

- Sponges excel at filtration (but so do oysters....) - however, I am not aware of any literature demonstrating that sponges have more efficient consumption than other animal lineages. Please provide a reference if available.

"Living sponges lack true tissues and organs, a successful adaptation for a filter-feeding lifestyle"

- That certainly depends on definition of tissues and organs, but many would allow calling pinacoderm tissue, and the pigment ring of many demosponge larvae an organ (see eg Leys et al 2002 for a description of this structure composed of several cell types).

Referee #2:

Review of Ruiz-Trillo, et al. Understanding animal origins

In this essay, Ruiz-Trillo and colleagues suggest an analogy between the technological development of the smartphone and the origin of animals. I am sympathetic to the aims of the manuscript and found the arguments about the origins of a 'new operating system' intriguing. But I found aspects under-developed and in some areas relying on outdated references. The comments below are provided with the aim of helping the authors sharpen their arguments.

As a rule, I make a habit of not suggesting authors refer to my own work. In this case, however, the authors have cited some of my older work and may not be aware of more recent papers which provide a different perspective.

Major concerns: My substantive concerns are with the discussion of the Ediacaran-Cambrian record, the failure to acknowledge relevant molecular clock data on divergences of early animal clades, which are critical to inferring evolutionary dynamics, and on the discussion of novelty/innovation.

The authors structure the paper as providing an 'alternative' to the view that extensive genetic novelty arose at the base of Metazoa. I guess that is fine, but there can't be many people interested in these issues who are not already aware of the magnificent work of the first author and his group (as well as the Sebe-Pedros and King groups) on Holozoa.

Initial models section -

That the apparent explosive origin of animals is 'misleading' has been well-established on both fossil and molecular clock evidence since at least 2011 and has been exhaustively discussed in dozens of papers. The Ediacaran soft-bodied biota precedes the Cambrian by tens of millions of years. Far from containing 'a paucity of fossils ... that could be related to living animals, Scott Evans (a former post-doc) and colleagues have shown that a number of Ediacaran clades represent metazoans. Kimberella has long been viewed as a lophotrochozoan, and one of the UK groups has also supported metazoan affinities of some Ediacaran fossils. Thus, just on fossil evidence the bilaterian divergence must pre-date 555 Ma. Wood et al. 2019 Nat Eco Evo and my 2020 Development review provide useful reviews, among others. The manuscript also fails to acknowledge extensive molecular clock estimates of metazoan divergences (although this was a major feature of the Erwin et al. 2011 paper, which is cited). See particularly the review paper by Stucky and Sperling. The recent Carlile et al. paper provides what I view as misleadingly young divergence estimates due to their calibration choices (I handled the paper at Science Advances, and was glad to see it published, but I don't agree with the estimates; but I no longer believe the estimates in Erwin et al. 2011, which are probably too old).

The 'in summary' claim at the bottom of page 3 has long been made by a variety of paleontologists, including Erwin and Valentine 2013.

Stepwise evolution model (pgs 4-5) - Brunet and King (2017) discuss the problems with the choano -> sponge transition at length (this paper is cited in the manuscript, although not here).

Pg 5, last full paragraph - while I agree with the points made here, most of those working on the ECR see the classic 'Cambrian explosion' as an episode affecting numerous independent bilaterian clades during which they acquired larger body size, hard

parts and a host of new ecological roles (including predation). But critically, these lineages seem to have predated the Cambrian radiation by perhaps 20 million years. I have argued in several papers that the difficulties, at least in part, stem from a failure to distinguish between the origins of evolutionary novelty (the acquisition of new characters) from innovation (the ecological and evolutionary success of clades containing evolutionary novelties) (see Erwin, Biological Reviews). Numerous cases in the fossil record have shown that these can be decoupled, contrary to the classic adaptive radiation model. Further, developmental novelties can potentiate later morphological novelties (see Erwin, J. Geological Society of London). On pg 13 these are conflated, which makes the logic unclear.

Minor concerns:

Pg 1, para 2 - 'both' animals?

Main text. The authors do not provide a definition of 'innovation' until the conclusion, which I feel is a mistake. The terms novelty and innovation have been used in so many ways (and often interchangeably) that I view it as essential that authors define how they are using the terms early in a paper.

Pg 4, top. "100 major eukaryotic divisions.. at the same taxonomic level.." What is the point here? Today we rely on phylogenies - it has been decades since most evolutionary biologists have viewed Linnean taxonomic ranks as relevant.

Pg 7, middle - I'm not sure why the technological combinatorics described here has to be 'sudden'. I quite agree with the point that both biological and technological advances often depend on the maturing of relevant attributes (this is a point I discuss as well in my forthcoming book on evolutionary novelty and innovation).

Pg 8 - I was surprised that transistors and integrated circuits were not included here, as they were arguably the key general-purpose technologies that enabled the aspects of technological innovation described here. They seem more akin to Holozoa.

Pg. 10 - How does the proposal here on the shift from temporal to spatially differentiated cell types differ from Mikhailov et al. 2009 and subsequent papers that built on these ideas? This section might also benefit from a more explicit fleshing out of the 'animal operating system' (is this just in GRNs?) and whether (as I suspect) there are significant enhancements in the OS at the last common bilaterian ancestor. Some discussion of the OS is on pg 12, but it seems a bit diffuse.

Much of this discussion would be improved by a more phylogenetic presentation - improving figure 2, and making clear what changes happen at which node. In phylogenetic trees nodes correspond with branches so Fig 2 needs additional branches inserted (leading to ghost taxa) for LUAA, FACA and LACA (others would use Metazoa rather than animals, but I suppose that is author's preference).

Pg. 13. Might be useful, if possible, to expand the discussion of co-option.

With these changes, this will be a most interesting contribution, which I look forward to seeing published in EMBO
Doug Erwin

Referee #3:

I enjoyed reading this paper, there's a lot of great content here and it is always fun to see what is on the minds of these esteemed authors. I think the idea is creative and out of the box. However, I'm not sure the smartphone analogy made things more clear for me. I personally found many of the components of the analogy to be a bit stretched / forced. For example, here is a summary sentence of the analogies section and what I was thinking as I read it:

"This can also be paralleled by the smartphone, which is a portable (motility), integrated (multicellular) computing device with multiple Apps (cell differentiation), input/output modalities (an opening), a user- friendly interactive display (the epithelium), and complex responsiveness (contractile capacity)."

Portability and motility seem different to me- the animal is actively motile, the smart phone is not. And it's not like the ancestor of animals was NOT motile or portable: protists swim, are carried on ocean currents, etc. The innovation is not motility, but a specific form of motility: fast, long-range actively powered motility, driven by muscles. This is a quantitative, not qualitative change, and it was only made possible by the evolution of large body size that changes the hydrodynamics of swimming through water. Phones, in contrast, arose from truly non-portable ancestors the size of buildings, and gradually shrunk enough to be carried around passively, much like the ancestor of animals presumably did on ocean currents. It seems to me that passive motility and a specific form of active motility are quite different, but I may be over thinking it.

How are 'integration' and 'cell differentiation' different here? I think of cell differentiation as being one of the most important parts of integration, myself.

Is cellular differentiation really like an app? I think of apps as being behavioral modules a phone can temporarily display, not structural components underpinning the way it is built (which is what I would argue cell differentiation is). In this analogy, an app would be more like the evolution of behaviors that can be switched over time, like contractile capacity, feeding, fleeing, mating, etc, right?

I'm not seeing how an epithelium is really analogous to a 'user friendly display', other than that it is specific type of surface material that is good at making a boundary. And even computers before smart phones have cases, which I think is an even better analogy for epithelium than a display is, in the sense that it is a boundary between the outside and inside that keeps sensitive internal components protected.

However, I liked the OS analogies, the transformative analogies, the diversification analogies, the historical contingency analogies, those all worked for me. Broadly, I think that this metaphor works at a high-level, but does not hold up well when made granular.

The Future Directions section makes points that I think overstates the impact of this analogy, and fundamentally changes the reader's perception of what the goal of this paper is.

"How might the smartphone analogy empower future research on the question of the origin of animals? First, it provides some concrete and testable predictions about the order of events between LUAA and FACA."

I don't really buy the argument that thinking about animals through the lens of smartphones will help us generate testable predictions about the origin of animals- the dynamics and mechanisms involved in each are totally different. To the extent that phones evolve, it is through cultural, not biological, mechanisms, with totally different underlying dynamics (it's based on human preferences, production costs, engineering advances, marketing capacity, profit motivations, etc. Phones are designed, animals evolve without a designer, etc). The differences in these processes are profound, which makes generating specific, testable predictions really difficult.

I also think that the specific components of the smartphone analogy are fairly weak, even if the overall idea is one that sort of makes sense. If one is using the analogy for scientific, not pedagogical purposes, this raises a set of philosophical questions: how do you know if your analogy is correct? Does correctness matter? If not, then how useful is it an analogy? How does the selection of alternative interpretations of the analogy change how your analogy creates scientific hypothesis? You should address these questions if you are using it for scientific purposes.

While I do agree with points made in the rest of the 'future directions' section, they aren't really conclusions of the smartphone metaphor. We've long known that animals evolved through the repurposing of ancestral modules, that the origin of animals created novel ecological feedback loops, etc. The literature on innovation more broadly has accommodated this thinking quite robustly at this point. I'd say that these important insights are metaphor agnostic- they are facts that are easily understood without metaphors.

This all being said, I am supportive of this paper for publication. I do not think my own somewhat luke-warm reception should preclude publication by any means. In general, I think it is good to have multiple ideas out there, and some bright graduate student might be sparked to think about major innovations in a new way as a result.

My advice, which is just that (not a requirement!), is to reframe this a little: make it clear that the analogy is mainly for pedagogical purposes, illustrating how innovations arise by using smartphones are something the reader has an intuitive understanding of. Make it clear that phones are not actually a great analogy for animals- they are so different in many ways- and yet, there are interesting commonalities that provide broader insight into the nature of transformative innovations. In short, I think it would be a more robust if you argue that the details do not actually matter and one should not focus too much on them, what matters is that it is a good teaching tool and that it illustrates general principles in an accessible manner.

Referee #4:

The authors provide an interesting opinion/idea about the analogy of the emergence of the smartphone with the emergence of the Metazoa. Since I have first heard about it I liked it. It is even very inspiring and can lead to new add-ons. E.g. epithelia and the recent case that became waterproof etc. As with all metaphors, there are the first inspiring aspects, but then the limitations of thought. So one should not take it too serious. It is also an example how technology impacts our view of nature. An older example were the "Kernels" for Gene regulatory networks.

The authors are very competent in evolutionary thinking and this is very good and visible in the manuscript. So I suggest the publication and have few remarks.

Abstract

The evolutionary transition from unicellular eukaryotes to animals occurred through a single evolutionary event.

this is difficult to tell, since it is misleading. It is still an accumulation of e.g. mutations, gene duplications etc. etc. while it seems on the tree it is a sudden appearance, we need to consider the lineage and not the node.

The diversity of animal life forms, both those currently living and those extinct, is astounding, yet all animals share a common ancestor (Medina et al. 2003; Ruiz-Trillo et al. 2008; Rokas 2008; Torruella et

al. 2012; Ros-Rocher et al. 2021).

please also cite older papers. the claim that all animals share a last common ancestor dates back more than than 100 years.

emergence of animals was a rare event, most likely involving some degree of contingency.

I guess we can assume that there were many unseccessful events and only one gave rise to the animals we see today. since we dont know how many e.g. ctenophore related lineages got extinct, it should be more carefully phrased.

since both of these events are singular: they happened just once in the history of life.

No, see comment above. Only because one lineage survived, we can not make this claim. There could have been many events. Since this is also not fossilized, we have no clue. It would be same to assume that the ctenophores feature emerged all at once. but this is obscured, because we lack a fossil record. and likely it was not the case.

Representatives of most animal phyla are found, including annelids (worms), arthropods (insects, crabs), molluscs (snails, octopuses), and chordates (including humans and fish) (Fedonkin, 2007).

the application of the artefactual term "phylum" (defined as "unique" body plan) is underlying the misconception of the cambrian explosion. see also Scholtz 2004 "Baupläne versus Ground Patterns, Phyla versus Monophyla: Aspects of Patterns and Processes in Evolutionary Developmental Biology" and Hejnol 2017 "Ladders, Trees, Complexity, and Other Metaphors in Evolutionary Thinking"

Therefore, and if you can not use clade or taxon instead of phylum, rephrase.

minor

(Ros-Rocher 2021)

coauthors missing..

If the multicellular stage persists long enough, the cells must adapt to this multicellular microenvironment, which includes biomechanical constraints different from those encountered in the unicellular stage. Under this condition, changes in the cytoplasm or cytoskeleton organization or even in the production of specific proteins (such as secretion or the positioning of specific adhesion molecules to the surface) could provide a specific phenotypic signature, initially environmentally induced, that could be reinforced due to prolonged exposure to this multicellular stage, inherited cortically, and even become fixed by epigenetic assimilation, eventually canalizing the evolution to a fixed multicellular entity (Sonnebornn 1970, Muroyama et al. 2023, Ashe et al. 2021, Weiner & Katz 2021).

There has been a recent manuscript on BioRxiv, which discusses a new cytoskeletal element:
<https://www.biorxiv.org/content/10.1101/2025.08.21.671432v1>

This raises the question of how we define an animal. If we discover an organism that branched off the tree between choanoflagellates and sponges/ctenophores (Fig. 1B) and is multicellular, should we call it a protist or an animal?

Do not forget the apomorphy of Metazoa: Spermatogenesis with sperm and oogenesis with a large egg. This should be the point of definition. (see e.g. Ax <https://link.springer.com/book/10.1007/978-3-642-80114-3>).

Reviewer 1

1) The current manuscript presents a origin of multicellular animals as a technological analogy. The review is well written and will likely be of interest to many readers of the E MBO journal. While I wholeheartedly agree that the smartphone analogy is very useful in conceptualizing the transition between coloniality and multicellularity, the manuscript contains views on sponge biology that are not necessarily correct - and importantly, are not key to the analogy used by the authors.

*We thank the reviewer for the positive feedback. We are particularly pleased that the smartphone analogy is found useful for conceptualizing this major evolutionary transition. We sincerely appreciate the reviewer for pointing out the inaccuracies in our discussion of sponge biology. We have carefully reviewed and corrected these sections, as detailed below.

2) "Although sponges may possess the simplest multicellular structure among extant animals"

- This is not true, Placozoans are much simpler, both in terms of overall morphology, size, and number of cell types. Consider giant barrel sponges or deep water carnivorous sponges and compare these to Trichoplax....

*We thank the reviewer for this insightful comment. The reviewer is correct that placozoans represent a simpler body plan than sponges. Our original statement was an oversimplification and we have now revised the sentence to be more precise: "Although

sponges may possess **one of** the simplest multicellular structure among extant animals,..”

3) "Third, the choanoblastea model was in part based on the perceived homology between an unusual cellular structure called the collar complex"

- The collar complex is not an unusual structure; in fact collar cells are present in all animals except Ctenophores and majority of Ecdysozoans - see Brunet and King 2017

*We thank the reviewer for this opportunity to clarify our phrasing. The reviewer is correct that the collar complex is widespread within Metazoa. Our use of the term “unusual” was intended to reflect its potential rarity across the broader eukaryotic tree of life, where it is, supposedly, a distinctive synapomorphy of the Choanozoa (the clade including animals and choanoflagellates). Indeed, as Brunet and King (2017) state in the paper cited by the reviewer: “the absence of a true collar complex from all non-choanozoans suggests that it is unlikely to evolve easily through convergence.” To avoid this ambiguity for our readers, we have replaced the word “unusual” and now describe the collar complex as a “**distinctive and evolutionarily conserved cellular structure**”. We also mention they are present in other metazoans.

4) However, recent morphological and molecular evidence suggests they arose independently by convergent evolution (Mah et al. 2014; Sogabe et al. 2019)."

- This is one viewpoint, but the issue is far from settled. See Peña et al 2016 for an alternative view, which should be considered and cited alongside those referred to by the authors.

*We thank the reviewer for the suggestion to present a more balanced view of this ongoing debate and pinpointing to this paper that we were not aware of. We have now revised the text and cite Peña et al. (2016). The revised sentence in the manuscript now reads: “*However, the evolutionary origin of this homology is debated. While some evidence suggests the collars of choanoflagellates and sponges arose independently by convergent evolution (Mah et al. 2014; Sogabe et al. 2019), other studies provide counter-evidence for a shared origin based on the deep conservation of the underlying molecular toolkit (Peña et al. 2016; Brunet & King 2017). Thus, the direct evolutionary lineage between these two cell types has been called into question.*”

5) "First, the further diversification of cell types eventually led to the formation of the first organs, including proto-muscles that could generate rapid, multicellular contractions, thereby maintaining motility as a key aspect of metazoan multicellularity."

- Many animals, especially aquatic larvae - but also adult flatworms and ctenophores - use cilia rather than muscle for movement. Given cilia were almost certainly present in the ancestors of animals, the first animals did not have to evolve muscle for movement.

*The reviewer is absolutely right to point out that ciliary-based locomotion (as well as others) is ancestral and that the first animals did not require muscle for motility. We have completely rewritten the sentence to frame the evolution of muscle as a subsequent innovation that enabled a new scale of movement.

The revised text now reads: *"First, the further diversification of cell types eventually led to the formation of the first organs. Among these were coordinated proto-muscles, **which enabled a new scale of rapid, multicellular contraction that added a more powerful way of locomotion to the metazoan repertoire.**"*

6) *"**In a similar way sponges might represent a "low-spec" version of multicellularity, retaining core features but shedding complexity for efficiency in certain, specific environments and lifestyles.**"*

- ***The authors' view that sponges are secondarily simplified is as valid as the alternative view - that their body plans represent ancestral condition for animals. Again, Trichoplax would be a "safer" example here.***

*We agree. We have changed the sentence to be more clear about this. The revised text now reads: *"In a similar way, animals like sponges or **placozoans** might represent a "low-spec" version of multicellularity, retaining core features but shedding complexity for efficiency in certain, specific environments and lifestyles. Sponges, for example, may have shed complex tissues and organs to become highly efficient at filter-feeding, thereby solving energy bottlenecks in specific environments (Asadzadeh et al. 2020). Nevertheless, **the alternative view that sponges instead may represent the ancestral body plan of animals remains a valid possibility.**"*

7) *"**Moreover, simplification often solves energy bottlenecks by focusing, for example, on more efficient consumption, something that sponges excel at.**"*

- ***Sponges excel at filtration (but so do oysters....) - however, I am not aware of any literature demonstrating that sponges have more efficient consumption than other animal lineages. Please provide a reference if available.***

*We thank the reviewer for this point and for prompting us to be more precise. We do not mean it has more efficient consumption than other animals, but that we argue that sponges seem to be exceptionally efficient at capturing food. This allows them to thrive in nutrient-poor environments. We have now clarified this in the text and provided a reference: **Asadzadeh** SS, Kiørboe T, Larsen P. S., Leys S. P., Yahel G., Walther J. H. (2020) Hydrodynamics of sponge pumps and evolution of the sponge body plan. eLife 9:e61012.

8) *"**Living sponges lack true tissues and organs, a successful adaptation for a filter-feeding lifestyle**"*

- ***That certainly depends on definition of tissues and organs, but many would allow calling pinacoderm tissue, and the pigment ring of many demosponge larvae an organ (see eg Leys et al 2002 for a description of this structure composed of several cell types).***

*We agree. We have changed the text to: *"**It has been proposed** that living sponges lack true tissues and organs, a successful adaptation for a filter-feeding lifestyle, yet they are built from the same fundamental molecular toolkit for multicellularity as more complex animal relatives."*

Reviewer 2

1) In this essay, Ruiz-Trillo and colleagues suggest an analogy between the technological development of the smartphone and the origin of animals. I am sympathetic to the aims of the manuscript and found the arguments about the origins of a 'new operating system' intriguing. But I found aspects under-developed and in some areas relying on outdated references. The comments below are provided with the aim of helping the authors sharpen their arguments.

As a rule, I make a habit of not suggesting authors refer to my own work. In this case, however, the authors have cited some of my older work and may not be aware of more recent papers which provide a different perspective.

Major concerns: My substantive concerns are with the discussion of the Ediacaran-Cambrian record, the failure to acknowledge relevant molecular clock data on divergences of early animal clades, which are critical to inferring evolutionary dynamics, and on the discussion of novelty/innovation.

The authors structure the paper as providing an 'alternative' to the view that extensive genetic novelty arose at the base of Metazoa. I guess that is fine, but there can't be many people interested in these issues who are not already aware of the magnificent work of the first author and his group (as well as the Sebe-Pedros and King groups) on Holozoa.

*We thank the reviewer. We are delighted the reviewer found the 'new operating system' analogy intriguing, and we agree that the manuscript can be improved by developing our arguments further. Regarding the overall framing, we agree that the pre-metazoan origin of the "animal toolkit" is a concept now well-established, but maybe not for the readers of EMBO Journal. Moreover, our intention is not to re-litigate this, but rather to propose a new conceptual framework—the smartphone analogy—to synthesize these findings and explore the process of complex innovation itself. We have revised the text to accommodate all the comments.

2)Initial models section -

That the apparent explosive origin of animals is 'misleading' has been well-established on both fossil and molecular clock evidence since at least 2011 and has been exhaustively discussed in dozens of papers. The Ediacaran soft-bodied biota precedes the Cambrian by tens of millions of years. Far from containing 'a paucity of fossils ... that could be related to living animals, Scott Evans (a former post-doc) and colleagues have shown that a number of Ediacaran clades represent metazoans. Kimberella has long been viewed as a lophotrochozoan, and one of the UK groups has also supported metazoan affinities of some Ediacaran fossils. Thus, just on fossil evidence the bilaterian divergence must pre-date 555 Ma. Wood et al. 2019 Nat Eco Evo and my 2020 Development review provide useful reviews, among others. The manuscript also fails to acknowledge extensive molecular clock estimates of metazoan divergences (although this was a major feature of the Erwin et al. 2011 paper, which is cited). See particularly

the review paper by Stucky and Sperling. The recent Carlile et al. paper provides what I view as misleadingly young divergence estimates due to their calibration choices (I handled the paper at Science Advances, and was glad to see it published, but I don't agree with the estimates; but I no longer believe the estimates in Erwin et al. 2011, which are probably too old). The 'in summary' claim at the bottom of page 3 has long been made by a variety of paleontologists, including Erwin and Valentine 2013.

*We thank the reviewer for this detailed feedback. Following his suggestions, we have substantially revised this paragraph. The new version now incorporates the critical context from molecular clock estimates, contrasts this with the fossil record, and discusses the co-option of gene regulatory networks as a key driver of the Cambrian radiation. We have added citations to Wood et al. (2019) and Erwin (2020) to support this more nuanced view.

The revised paragraph reads as: *“This concept of the explosive origin of animals has been strengthened by the apparent sparsity of fossils in the pre-Cambrian Ediacaran period that could be related to living animals or Cambrian fossils. However, to counter this view, some have argued that the Cambrian explosion is an artefact of the fossil record, and similar animals could have originated long before but remain undetected because they did not fossilise well, probably due to their small size or the lack of hard body parts (Erwin and Valentine, 2013). Moreover, recent analyses of the fossil record and molecular clock estimates -which consistently place the origin of Metazoa nearly 200 million years before the Cambrian- have further challenged the explosive origin of animals, implying a long history of cryptic evolution (Erwin 2020, Wood et al. 2019) . Instead authors have proposed the early metazoan diversification was indeed the result of successive, transitional radiations that originated on the late Ediacaran, being the Cambrian explosion a radiation of crown-group bilaterians (Wood et al. 2019). Thus, the concept of the "Cambrian explosion" has been reinterpreted. It is now seen not as an explosion of genetic innovation, but as a rapid diversification of bilaterian body plans. This radiation was likely fueled by a combination of factors, including environmental changes (e.g., rising oxygen levels), new ecological pressures (e.g., a predator-prey "arms-race"), and, probably, the co-option of ancient gene regulatory networks into new roles for body plan patterning. (Erwin and Valentine, 2013)”*

3) Stepwise evolution model (pgs 4-5) - Brunet and King (2017) discuss the problems with the choano -> sponge transition at length (this paper is cited in the manuscript, although not here).

*Thank you. Fixed.

4) Pg 5, last full paragraph - while I agree with the points made here, most of those working on the ECR see the classic 'Cambrian explosion' as an episode affecting numerous independent bilaterian clades during which they acquired larger body size, hard parts and a host of new ecological roles (including predation). But critically, these lineages seem to have predated the Cambrian radiation by perhaps 20 million years. I have argued in several papers that the difficulties, at least in part, stem from a failure to distinguish between the origins of evolutionary novelty (the acquisition of new

characters) from innovation (the ecological and evolutionary success of clades containing evolutionary novelties) (see Erwin, Biological Reviews). Numerous cases in the fossil record have shown that these can be decoupled, contrary to the classic adaptive radiation model. Further, developmental novelties can potentiate later morphological novelties (see Erwin, J. Geological Society of London). On pg 13 these are conflated, which makes the logic unclear.

*Thank you. We agree that evolutionary novelties can be decoupled from the real innovation and this is indeed an important aspect. Following this advice, we now use these precise terms to clarify our argument: the origin of several biological technologies represents the key novelty, while the subsequent radiation represents the successful innovation with the “animal OS”.

5) Pg 1, para 2 - 'both' animals?

*Thank you. Fixed. It now reads “The diversity of animal life, encompassing both extant and extinct forms, is... “

6) Main text. The authors do not provide a definition of 'innovation' until the conclusion, which I feel is a mistake. The terms novelty and innovation have been used in so many ways (and often interchangeably) that I view it as essential that authors define how they are using the terms early in a paper.

*We agree that the term “innovation” should have been defined at the beginning of the paper, given its central role in our narrative. We have now added a short definition within the introductory paragraph:

*“This suggests that at some point in evolution, a single-celled ancestor of animals (known as the last unicellular ancestor of animals (LUAA)) acquired **some innovation —that is, a novel trait or capacity that created new functional or ecological possibilities (Erwin 2015) that subsequently enabled the formation of the first animal common ancestor (FACA).**”*

We have also added an additional reference accordingly:

Erwin, D. H. (2015). Novelty and innovation in the history of life. *Current Biology*, 25(19), R930–R940.

7) Pg 4, top. "100 major eukaryotic divisions.. at the same taxonomic level.." What is the point here? Today we rely on phylogenies - it has been decades since most evolutionary biologists have viewed Linnean taxonomic ranks as relevant.

*We agree. Our intention was to highlight that animals represent just one of many deep-branching eukaryotic lineages, in order to contextualize their place within the broader

eukaryotic tree of life. Those 100 major eukaryotic lineages are indeed based on phylogeny.

We have revised the text and now it reads as *“However, the reality is that animals constitute only one of approximately 100 major, deep-branching eukaryotic lineages (del Campo et al. 2014).”*

8) Pg 7, middle - I'm not sure why the technological combinatorics described here has to be 'sudden'. I quite agree with the point that both biological and technological advances often depend on the maturing of relevant attributes (this is a point I discuss as well in my forthcoming book on evolutionary novelty and innovation).

*We agree. We deleted “suddenly”. The sentence now reads *“These technologies matured over time until they were integrated into a single device.”*

9) Pg 8 - I was surprised that transistors and integrated circuits were not included here, as they were arguably the key general-purpose technologies that enabled the aspects of technological innovation described here. They seem more akin to Holozoa.

*We thank the reviewer for this interesting point and how it can reinforce our analogy between IT and the emergence of holozoans. We do included these elements (implicitly) within the “microchip” item that appears in our comparative diagram (animals/smartphones). But the referee is right in that this should be highlighted, since transistors are the crucial component of the IT era. We have added a sentence to the first paragraph in the section on IT and the Holozoan revolution:

“The rise of the IT era (thanks to the invention of the transistor, which allowed the building integrated circuits) marks a rapid progression in inventions related to digital systems, new materials, and networking.”

And we have also followed the referee’s suggestion of making the analogy between the rise of integrated circuits and the Holozoan revolution, adding a sentence at the end of the second paragraph in the same section: *“In our technological analogy, the beginning of the Holozoan Revolution parallels the invention of the transistor, followed by the swift proliferation of integrated circuit architectures.”*

10) Pg. 10 - How does the proposal here on the shift from temporal to spatially differentiated cell types differ from Mikhailov et al. 2009 and subsequent papers that built on these ideas? This section might also benefit from a more explicit fleshing out of the 'animal operating system' (is this just in GRNs?) and whether (as I suspect) there are significant enhancements in the OS at the last common bilaterian ancestor. Some discussion of the OS is on pg 12, but it seems a bit diffuse.

* We thank the reviewer for pushing us to clarify these two central points. The key difference from us and Mikhailov et al. (2009) is the timing and nature of the temporal-

to-spatial transition. The Mikhailov et al. model places this transition as the key event in the origin of animals (Metazoa). Our model, informed by recent discoveries of spatial differentiation and spatial cell differentiation mechanisms in unicellular relatives, proposes that a basic capacity for spatial organization of cell types already existed in the pre-metazoan Last Unicellular Ancestor of Animals (LUAA). We have now clarified this in the text. With regards the "Animal OS". We see the "Animal OS" as more than just Gene Regulatory Networks. We define it as the hierarchical integration of multiple pre-existing systems, including: Gene Regulatory Networks (GRNs) that define cell identity, Cell-cell signaling and adhesion pathways and mechanisms to control cell number and stoichiometry. Regarding the reviewer's suspicion about enhancements in bilaterians, we agree completely. We now explicitly propose that the evolution of more complex gene regulatory networks or chromatin architecture (e.g., distal enhancers and TADs) in the last common bilaterian ancestor represents a major "OS upgrade" that enabled more complex body plans. We have now integrated these concepts in a more clear way into the manuscript.

11) Much of this discussion would be improved by a more phylogenetic presentation - improving figure 2, and making clear what changes happen at which node. In phylogenetic trees nodes correspond with branches so Fig 2 needs additional branches inserted (leading to ghost taxa) for LUAA, FACA and LACA (others would use Metazoa rather than animals, but I suppose that is author's preference).

* We agree. We have made changes to Figure 2 following the reviewer's suggestions.

12) Pg. 13. Might be useful, if possible, to expand the discussion of co-option.

* We have expanded a little bit the discussion of co-option with a couple of examples.

Reviewer 3

1) I enjoyed reading this paper, there's a lot of great content here and it is always fun to see what is on the minds of these esteemed authors. I think the idea is creative and out of the box. However, I'm not sure the smartphone analogy made things more clear for me. I personally found many of the components of the analogy to be a bit stretched / forced. For example, here is a summary sentence of the analogies section and what I was thinking as I read it:

"This can also be paralleled by the smartphone, which is a portable (motility), integrated (multicellular) computing device with multiple Apps (cell differentiation), input/output modalities (an opening), a user- friendly interactive display (the epithelium), and complex responsiveness (contractile capacity)."

Portability and motility seem different to me- the animal is actively motile, the smart phone is not. And it's not like the ancestor of animals was NOT motile or portable:

protists swim, are carried on ocean currents, etc. The innovation is not motility, but a specific form of motility: fast, long-range actively powered motility, driven by muscles. This is a quantitative, not qualitative change, and it was only made possible by the evolution of large body size that changes the hydrodynamics of swimming through water. Phones, in contrast, arose from truly non-portable ancestors the size of buildings, and gradually shrunk enough to be carried around passively, much like the ancestor of animals presumably did on ocean currents. It seems to me that passive motility and a specific form of active motility are quite different, but I may be over thinking it.

*We thank the reviewer for the positive assessment and for the deep and thoughtful engagement with our smartphone analogy. We are delighted the reviewer found the idea creative, and the detailed feedback is exactly the kind of critical thinking we had hoped to inspire. This comment made us realize that the metaphor could be misinterpreted as a direct literal comparison. For a metaphor to be scientifically useful it needs to be used productively. The point is not to have perfect correspondence between the phenomena and the metaphor, since that will produce a replica of the phenomena and that will not be interesting and will not have any predictive power. We have now included a couple of disclaimers in the text and an explanation of the power of metaphors in science (see also other answers below).

The new paragraph reads as this: *“We begin by reviewing current models for the origin of animals and examining how recent evidence challenges them. We then turn to a powerful, if sometimes misunderstood, tool of scientific inquiry: the analogy. Analogies in science are not rhetorical flourishes but instruments of discovery—they can spark new hypotheses by revealing unseen connections between domains. For an analogy to be scientifically fruitful, it must have three components: the positive analogy (known similarities that ground the comparison), the negative analogy (known differences that define its limits), and, most importantly, the neutral analogy (the unknown properties where the comparison might or might not hold). It is by exploring this “neutral analogy” that new, testable hypotheses are generated. We here use the evolution of the smartphone as an analogy for the origin of animals. We invite the reader to engage with this comparison in the spirit just outlined: to recognize and/or challenge the positive analogies, to acknowledge the negative ones, and to challenge and explore the neutral ones that may inspire fresh thinking. Through this metaphor, our goal is not to be pedagogical but rather to challenge old assumptions and generate new questions about one of life’s major evolutionary transitions..”*

We agree with the reviewer that animal motility is (mostly) active while smartphone portability is passive. We also agree protists were motile. However, our intention was not to draw a direct equivalence between the mechanisms of movement, but rather to highlight the emergence of an integrated, multi-functional system that gains a new degree of freedom or motility.

We consider the coordinated metazoan motility to be an important innovation. Similarly, before the smartphone, powerful computing was largely fixed in place. The key innovation was not just portability (a book is portable), but the portability of a complex, integrated computing system. In both cases, a new level of integrated complexity achieved freedom of movement, which had profound consequences. We made some changes to the motility part in other sections (see previous answer) that we hope will make this more clear.

2) How are 'integration' and 'cell differentiation' different here? I think of cell differentiation as being one of the most important parts of integration, myself.

*We agree that both integration and cell differentiation are very related. In our model, we see integration as the process of coordinating pre-existing molecular systems (e.g., adhesion, signaling, and gene regulatory networks) into a cohesive, functional whole. We see spatial cell differentiation (i.e., the division of labor) as a key emergent property that is made possible by that integration and/or that requires that integration.

3) Is cellular differentiation really like an app? I think of apps as being behavioral modules a phone can temporarily display, not structural components underpinning the way it is built (which is what I would argue cell differentiation is). In this analogy, an app would be more like the evolution of behaviors that can be switched over time, like contractile capacity, feeding, fleeing, mating, etc, right?

*We totally agree and we thank the reviewer. We agree that equating "cell differentiation" directly with an "app" is a flawed parallel, for the reasons outlined. Our intended—and more robust—analogy, which we use elsewhere in the manuscript, is that "apps" represent larger, integrated functional modules like new body plans or ecological strategies. We have now revised that sentence to be consistent with our vision of "apps". The sentence now says "...computing device with multiple Apps (**body plans, ecological strategies**).."

4) I'm not seeing how an epithelium is really analogous to a 'user friendly display', other than that it is specific type of surface material that is good at making a boundary. And even computers before smart phones have cases, which I think is an even better analogy for epithelium than a display is, in the sense that it is a boundary between the outside and inside that keeps sensitive internal components protected.

*We thank the reviewer for this suggestion. A phone 'case' is a much more fitting analogy for an epithelium than a 'display,' as its primary role is to create a protective boundary for sensitive internal components. Again, this is what we were looking for with this metaphor, to create discussion about the different levels of the analogy. We now incorporate cases in the analogy.

5) However, I liked the OS analogies, the transformative analogies, the diversification analogies, the historical contingency analogies, those all worked for me. Broadly, I think that this metaphor works at a high-level, but does not hold up well when made granular.

*We thank the reviewer for this positive overall assessment. We also agree with the point that the metaphor is strongest at its conceptual level and can become strained when the granular comparisons are too literal. As explained above, the idea is not to have perfect correspondence between the phenomena and the metaphor, but rather to have a metaphor that is productive and can produce testable hypotheses. All this is now made explicit in the introduction and discussion.

6) The Future Directions section makes points that I think overstates the impact of this analogy, and fundamentally changes the reader's perception of what the goal of this paper is.

"How might the smartphone analogy empower future research on the question of the origin of animals?

First, it provides some concrete and testable predictions about the order of events between LUAA and FACA."

I don't really buy the argument that thinking about animals through the lens of smartphones will help us generate testable predictions about the origin of animals- the dynamics and mechanisms involved in each are totally different. To the extent that phones evolve, it is through cultural, not biological, mechanisms, with totally different underlying dynamics (it's based on human preferences, production costs, engineering advances, marketing capacity, profit motivations, etc. Phones are designed, animals evolve without a designer, etc). The differences in these processes are profound, which makes generating specific, testable predictions really difficult.

*We thank the reviewer for raising these important philosophical questions about the scientific utility of our analogy. This is a crucial point, and we appreciate the opportunity to clarify the methodological foundation for our approach.

To address these excellent questions, we frame our analogy within the formal structure for the scientific use of models proposed by philosopher of science Mary Hesse in her book "*Models and Analogies in Science*". Hesse argued that any scientific analogy consists of three parts:

-The Positive Analogy: These included the shared properties that make the comparison valid in the first place. These are the high-level patterns such as the integration of pre-existing modules.

-The Negative Analogy: The known differences, which are crucial to acknowledge. We agree with the reviewer that there are profound differences, such as intentional design versus natural selection (which we do mention in the manuscript). These negative analogies are very much needed and define the clear limits of the comparison.

-The Neutral Analogy: The properties where it is unknown if the two systems are similar. As Hesse argued, the scientific power of an analogy lies in its **neutral analogy**, which serves as an engine for generating new, testable hypotheses. This directly addresses the reviewer's questions about correctness and utility. The value of our analogy is not its literal "correctness" (which is impossible, given the negative analogy), but its productivity in turning points in the neutral analogy into testable questions.

For example, the neutral analogy prompts us to ask: "If technological evolution proceeds through stable, intermediate prototypes (like PDAs before smartphones), did animal evolution also involve stable, intermediate forms before the emergence of the final body plans?" This leads directly to our testable prediction about the existence of organisms

that fit our LUAA criteria (stable, differentiated multicellularity) but lack the full FACA "OS" (e.g., true epithelial polarity).

To clarify this, we have revised the manuscript to briefly frame our analogy within this context, and acknowledge its role as a heuristic tool for generating hypotheses from its "neutral" components. See also answer to point 1.

7) I also think that the specific components of the smartphone analogy are fairly weak, even if the overall idea is one that sort of makes sense. If one is using the analogy for scientific, not pedagogical purposes, this raises a set of philosophical questions: how do you know if your analogy is correct? Does correctness matter? If not, then how useful is it an analogy? How does the selection of alternative interpretations of the analogy change how your analogy creates scientific hypothesis? You should address these questions if you are using it for scientific purposes.

* This is a crucial point, and we believe we have answered it in the point above. The value of the metaphor lies not in being literally correct, but in being scientifically productive. The new section describing how we expect the analogy to work should clarify all this.

8) While I do agree with points made in the rest of the 'future directions' section, they aren't really conclusions of the smartphone metaphor. We've long known that animals evolved through the repurposing of ancestral modules, that the origin of animals created novel ecological feedback loops, etc. The literature on innovation more broadly has accommodated this thinking quite robustly at this point. I'd say that these important insights are metaphor agnostic- they are facts that are easily understood without metaphors.

*We agree entirely that foundational concepts like the repurposing of ancestral modules and the emergence of novel ecological feedback are well-established principles in modern evolutionary biology.

Our goal was not to claim these principles as novel conclusions derived from the metaphor. Rather, we use the analogy as a synthesizing framework and a way to test what are parallels (analogy) that work (positive) and the ones that do not work (negative). Therefore, while some of the insights mentioned by the reviewer are indeed 'metaphor agnostic' when considered individually, the analogy provides a tool to understand them as an interconnected whole and to structure our thinking about their interplay during a major evolutionary transition.

9) This all being said, I am supportive of this paper for publication. I do not think my own somewhat luke-warm reception should preclude publication by any means. In general, I think it is good to have multiple ideas out there, and some bright graduate student might be sparked to think about major innovations in a new way as a result.

My advice, which is just that (not a requirement!), is to reframe this a little: make it clear that the analogy is mainly for pedagogical purposes, illustrating how innovations arise by using smartphones are something the reader has an intuitive understanding of.

Make it clear that phones are not actually a great analogy for animals- they are so different in many ways- and yet, there are interesting commonalities that provide broader insight into the nature of transformative innovations. In short, I think it would be a more robust if you argue that the details do not actually matter and one should not focus too much on them, what matters is that it is a good teaching tool and that it illustrates general principles in an accessible manner.

*We thank the reviewer for the support. We believe the explanation on positive, negative and neutral analogies should now clarify what we are looking for, which is more than a pedagogical metaphor. The fact that animals and phones are so different in so many ways while having interesting commonalities is what make, for us, this metaphor so powerful.

Reviewer 4

1) The authors provide an interesting opinion/idea about the analogy of the emergence of the smartphone with the emergence of the Metazoa. Since I have first heard about it I liked it. It is even very inspiring and can lead to new add-ons. E.g. epithelia and the recent case that became waterproof etc. As with all metaphors, there are the first inspiring aspects, but then the limitations of thought. So one should not take it too serious. It is also an example how technology impacts our view of nature. An older example were the "Kernels" for Gene regulatory networks.

The authors are very competent in evolutionary thinking and this is very good and visible in the manuscript. So I suggest the publication and have few remarks.

*We thank the reviewer for the positive assessment and for the thoughtful engagement with our analogy. We are delighted the reviewer found the analogy inspiring and we particularly enjoyed the parallel between epithelia and waterproof cases!

2) Abstract

The evolutionary transition from unicellular eukaryotes to animals occurred through a single evolutionary event.

this is difficult to tell, since it is misleading. It is still an accumulation of e.g. mutations, gene duplications etc. etc. while it seems on the tree it is a sudden appearance, we need to consider the lineage and not the node.

*We agree and we have changed the sentence to make the point more clear, which now reads as "How animals evolved from their unicellular ancestor is a fundamental biological question. The fact that all animals are monophyletic—sharing a single common ancestor—implies their origin from unicellular eukaryotes was likely driven by rare and highly advantageous innovations. "

3) The diversity of animal life forms, both those currently living and those extinct, is astounding, yet all animals share a common ancestor (Medina et al. 2003; Ruiz-Trillo et al. 2008; Rokas 2008; Torruella et al. 2012; Ros-Rocher et al. 2021).

please also cite older papers. the claim that all animals share a last common ancestor dates back more than 100 years.

*We agree. Our original intention was to cite the modern molecular work that definitively confirmed the monophyly of animals. However, the reviewer is absolutely right that we should also acknowledge the foundational, pre-molecular origins of this idea. Following the reviewer's advice we now cite Haeckel 1874; and also Wainright et al. 1993 as one of the pioneers in the molecular work.

4) emergence of animals was a rare event, most likely involving some degree of contingency.

I guess we can assume that there were many unseccessful events and only one gave rise to the animals we see today. since we dont know how many e.g. ctenophore related lineages got extinct, it should be more carefully phrased.

*We totally agree!. We now say "extant animals"

5) since both of these events are singular: they happened just once in the history of life. No, see comment above. Only because one lineage survived, we can not make this claim. There could have been many events. Since this is also not fossilized, we have no clue. It would be same to assume that the ctenophores feature emerged all at once. but this is obscured, because we lack a fossil record. and likely it was not the case.

*Again, we totally agree!, and we thank the reviewer for this. we have now changed this sentence to "...since both of these events are singular, in each case there is only one extant lineage, demonstrating acquisition of innovations that led to highly successful descendants."

6) Representatives of most animal phyla are found, including annelids (worms), arthropods (insects, crabs), molluscs (snails, octopuses), and chordates (including humans and fish) (Fedonkin, 2007).

the application of the artefactual term "phylum" (defined as "unique" body plan) is underlying the misconception of the cambrian explosion. see also Scholtz 2004 "Baupläne versus Ground Patterns, Phyla versus Monophyla: Aspects of Patterns and Processes in Evolutionary Developmental Biology" and Hejnol 2017 "Ladders, Trees, Complexity, and Other Metaphors in Evolutionary Thinking" Therefore, and if you can not use clade or taxon instead of phylum, rephrase.

*We totally agree and we now use "clades".

Dear Prof. Ruiz-Trillo,

As you will see below, all four referees are very positive about your revised manuscript. There are a couple of minor issues remaining that I believe you can quickly revise. Additionally, there are a few more technical issues that were picked by our editorial assistance team that you can find below and I will ask you to fix them so we can proceed with official acceptance of your manuscript.

One last point: we allow (and even encourage) including in our review articles glossary as well as boxes. The glossary can be very useful for readers outside of the field and the boxes can be helpful in highlighting in more detail a specific subject that including it in detail in the main text can somewhat disrupt the flow. While it is not mandatory in any way to use these features, I wanted to bring them to your attention and to provide you with the opportunity to incorporate these into your manuscript, should you find it useful.

Please try to send us your revised manuscript at your earliest convenience.

Thank you for your scientific contribution with this manuscript. I look forward to your final revision.

Yours sincerely,

Yehu Moran
Academic Editor
The EMBO Journal

We realize that it is difficult to revise to a specific deadline. In the interest of protecting the conceptual advance provided by the work, we recommend a revision within 3 months (9th Feb 2026). Please discuss the revision progress ahead of this time with

the editor if you require more time to complete the revisions.

Specific comments by editorial assistance team

Keywords: missing - Please add up to five keywords.

AUTHORS: Please define the corresponding author on the manuscript title page and add the email address.

REFERENCE FORMAT: Please list up to 10 authors before et al. - last name, followed by the initials of the first name(s) - and remove any dois for published works. E.G. see Urrutia et al (2021).

Referee reports

Referee #1:

I am happy to see changes in the manuscript following referees' comments, including my own. I believe it is now suitable for publication.

Referee #2:

The authors have substantially modified the manuscript, and I think much improved it in the process.

My only remaining comment is that Simon Conway Morris's last name is Conway Morris, not Morris, so he should appear in the refs as Conway Morris (yes, others have cited him as Morris, S.C., but I've known him for decades, and his last name is Conway Morris (no hyphen).

Referee #3:

I am happy with this version. It's not the paper I would have written, but thankfully that is not a criterion for publication! I think they addressed my concerns pretty well without having to do a fundamental rewrite. I support publication of this version.

Referee #4:

The manuscript has improved, I have only one last remark:

As a response to a reviewers 1 comment the authors state now:

"the alternative view that sponges instead may represent the ancestral body plan of animals remains a valid possibility."

Here is a common misconception of animal evolution that the tip of a lineage represents an ancestral state. That sponges (or ctenophores) represent an ancestral body plan is very unlikely and neglects that evolution happens in all lineages. So it gives a wrong message. You can replace the sentence with "or not": "Sponges, for example, may or may not have shed complex tissues and organs to become highly efficient at filter-feeding, thereby solving energy bottlenecks in specific environments (Asadzadeh et al. 2020)".

Reviewer 1

1) I am happy to see changes in the manuscript following referees' comments, including my own. I believe it is now suitable for publication.

*Thanks so much!

Reviewer 2

1) The authors have substantially modified the manuscript, and I think much improved it in the process.

My only remaining comment is that Simon Conway Morris's last name is ConwayMorr is, not Morris, so he should appear in the refs as Conway Morris (yes, others have cited him as Morris, S.C., but I've known him for decades, and his last name is ConwayMorr is (no hyphen).

The authors structure the paper as providing an 'alternative' to the view that extensive genetic novelty arose at the base of Metazoa. I guess that is fine, but there can't be many people interested in these issues who are not already aware of the magnificent work of the first author and his group (as well as the Sebe-Pedros and King groups) on Holozoa.

*Thanks so much! And thank you for spotting this! We were indeed aware that Simon's last name is Conway Morris, but this error unfortunately escaped our notice. We have corrected it in the references.

Reviewer 3

1) I am happy with this version. It's not the paper I would have written, but thankfully that is not a criterion for publication! I think they addressed my concerns pretty well without having to do a fundamental rewrite. I support publication of this version.

*Thanks so much! We appreciate the reviewer's support and his/her open-mindedness regarding our approach!

Reviewer 4

1) The manuscript has improved, I have only one last remark:

As a response to a reviewers 1 comment the authors state now:

"the alternative view that sponges instead may represent the ancestral body plan of animals remains a valid possibility."

Here is a common misconception of animal evolution that the tip of a lineage represents an ancestral state. That sponges (or ctenophores) represent an ancestral body plan is very unlikely and neglects that evolution happens in all lineages. So it gives a wrong message. You can replace the sentence with "or not": "Sponges, for example, may or may not have shed complex tissues and organs to become highly efficient at filter-feeding, thereby solving energy bottlenecks in specific environments (Asadzadeh et al. 2020)".

*Thanks so much! We totally agree and this is not the message we wanted to convey. In principle, yes, it is a valid possibility, but also it may not. We have edited this. For example when mentioning that "Although sponges may possess one of the simplest multicellular structure among extant animals, we lack definitive evidence that the first animal resembled them.", we now include this other sentence: "Indeed, this is quite unlikely given the time span that has occurred since they last diverged from other animals." On the section mentioned by the reviewer, we have made the changes suggested and now reads as "Sponges, for example, may or may not have shed complex tissues and organs to become highly efficient at filter-feeding, thereby solving energy bottlenecks in specific environments (Asadzadeh et al. 2020). Nevertheless, the alternative view that sponges instead may represent the ancestral body plan of animals remains an unlikely but valid possibility."

Dear Prof. Ruiz-Trillo,

I am pleased to inform you that your manuscript has been accepted for publication in the EMBO Journal.

Your manuscript will be processed for publication by EMBO Press. It will be copy edited and you will receive page proofs prior to publication. Please note that you will be contacted by Springer Nature Author Services to complete licensing and payment information. As this review article was invited by us, we will provide you in a separate email with a waiver token you can put into the Springer-Nature system to fully cover the publication costs.

Yours sincerely,

Yehu Moran
Academic Editor
The EMBO Journal

Please note that it is The EMBO Journal policy for the transcript of the editorial process (containing referee reports and your response letters) to be published as an online supplement to each paper. If you should prefer removal of any referee-only figures included in the point-by-point response(s), e.g. because they may still be used for future publication or because they have been reproduced from published work by others, please do let us know immediately via response email.

More information is available here: <https://link.springer.com/partners/embo-press/editorial-policies#Peer%20review>